# PROVABLY ACCURATE ODE FORECASTING THROUGH EXPLICIT TRAJECTORY OPTIMIZATION

## ABSTRACT

This work introduces a method to enable accurate forecasting of time series governed by ordinary differential equations (ODE) through the usage of cost functions explicitly dependent on the future trajectory rather than the past measurement times. We prove that the space of solutions of an $N$-dimensional, smooth, Lipschitz ODE on any given finite time horizon is an $N$-dimensional Riemannian manifold embedded in the space of square integrable continuous functions. This finite dimensional manifold structure enables the application of common statistical objectives such as maximum likelihood (ML), maximum a posteriori (MAP), and minimum mean squared error (MMSE) estimation directly in the space of feasible ODE solutions. The restriction to feasible trajectories of the system limits known issues such as oversmoothing seen in unconstrained MMSE forecasting. We demonstrate that direct optimization of trajectories reduces error in forecasting when compared to estimating initial conditions or minimizing empirical error. Beyond theoretical justifications, we provide Monte Carlo simulations evaluating the performance of the optimal solutions of six different objective functions: ML, MAP state estimation, MMSE state estimation, MAP trajectory estimation, MMSE trajectory estimation over all square integrable functions, and MMSE trajectory estimation over solutions of the differential equation.

## 1 INTRODUCTION

Decision making often hinges on accurate forecasts. Due to the simplicity of finite-dimensional spaces, many problems in state estimation and forecasting are formulated in a pointwise manner over time. Despite this computational simplification, feasible trajectories must solve consistency constraints, and additionally pointwise estimation may be unnecessary because the states at different points in time are often highly correlated. Furthermore, the structure of the trajectory itself often contains meaningful information beyond that of the value at a particular time horizon. Time series forecast consistency is often enforced after the fact, where predictions for distinct time horizons are constructed and then projected to enforce the hierarchical constraint (Rangapuram et al., 2021; 2023).

In this work, we forecast ordinary differential equations (ODEs) over an entire chosen time horizon by formulating the time series forecasting problem as a finite-dimensional point estimation problem on a Riemannian manifold. We prove that the space of feasible trajectories on any finite time-interval of a smooth Lipschitz dynamical system is itself an $N$-dimensional Riemannian submanifold of the space of continuous bounded functions, where $N$ is the dimensionality of the state. We argue that neural ODEs (Chen et al., 2018) and related differential equation modeling methods(Dupont et al., 2019; Greydanus et al., 2019; Massaroli et al., 2020; Finlay et al., 2020; Biloš et al., 2021; Holt et al., 2022) fundamentally solve a problem of point estimation in the manifold largely without consideration of the statistical distinction between parameter estimation and trajectory estimation. We then use this observation to cast the trajectory estimation problem into a classical statistical framework, enabling direct optimization of statistical objectives based on forecasting. Furthermore, we provide tractable computational methods to optimize these objectives.

The rest of the manuscript is organized as follows. In Section 2, we introduce the model formulation, assumptions, and key spaces in this work. In Section 3 we prove the existence of the finite-dimensional Riemannian trajectory manifold. Then, in Section 4, we describe the tools required for statistical estimation on the trajectory manifold based on noisy measurements, as well as a description of the

implications for commonly used statistical objectives. In Section 5, we provide computational tools for optimizing these common statistical objectives directly on the manifold of valid trajectories. Finally, in Section 6, we provide numerical simulations that demonstrate that the proposed explicit trajectory optimization outperforms the standard data fitting objective.

## 1.1 RELATED WORK

**Differential Equations and Deep Learning**   The connection between deep learning and differential equations can be broadly partitioned into two major categories. In physics-informed machine learning, neural networks are used to approximately solve a known differential equation subject to noisy observations. On the other side, there has been significant interest in using neural networks to learn a representation of the underlying differential equation.

While using neural networks as solutions to differential equations dates back to at least the 1990's (Lagaris et al., 1998), it has had a significant resurgence in recent years (Raissi & Karniadakis, 2018; Raissi et al., 2019). Commonly, these techniques use some form of empirical risk minimization, an objective which is closely related to maximum likelihood estimation in classical estimation and represents a notion of best fit to the observed data. In these techniques, rather than use the differential equation as a hard constraint, it is used as a regularization for regression. While this regularization approach has a number of desirable characteristics — simple optimization and an automatic tolerance for input perturbations to the system — it can result in less readily interpretable behavior. Despite these limitations, the regularization suggests the ability to use the high level of structure in the space of differential equation solutions for inference.

From a different direction, numerous techniques around continuous-time machine learning such as Neural Ordinary Differential Equations (ODEs) have shown promise in time-series forecasting when the dynamics are unknown (Chen et al., 2018). In such techniques, a neural network is used to learn an approximation of the underlying differential equation. It was quickly noted that ODE solutions have fundamental topological constraints, and so augmented Neural ODEs were proposed (Dupont et al., 2019). Further analysis of the underlying behavior of ODE-defined models led to the construction of different time-varying versions of the model, as well as the usage of data-dependent vector fields (Massaroli et al., 2020). While there have been numerous application-focused papers using neural ODEs (Chen et al., 2022), many of the additional advancements have been in methods of training the models (Finlay et al., 2020). Alternative perspectives have been explored in searching for solutions in the Laplace domain (Holt et al., 2022). A subtle change in approach was used in Neural Flows, which in principle operate in the space of trajectories instead of the dynamics, but does so through a restriction of a set of functions satisfying some necessary conditions of flows (Biloš et al., 2021).

**Time Series Forecasting Regularization**   As regularization is an essential part of ensuring models are generalizable, we include a brief summary of key time series forecasting regularization methods.

Deep learning models have many universal approaches to regularization which are independent of the context. Some common approaches include dropout (Srivastava et al., 2014), dropconnect (Wan et al., 2013), batch normalization (Ioffe & Szegedy, 2015), complexity regularization (Barron, 1991), L0 regularization (Louizos et al., 2018), and classical regularizers such as Tikhonov regularization and LASSO.

Beyond general techniques, time series forecasting necessitates specialized techniques due in part to the lack of independent samples. There exist numerous specialized regularization methods for recurrent neural networks (Zaremba et al., 2014; Krueger & Memisevic, 2016; Wang & Niepert, 2019; Krueger et al., 2017). Neural ODEs introduce additional complexities in the training process, and have thus spawned a number of specific regularization techniques. These include random integration times (Ghosh et al., 2020), penalties based on the Jacobian of the vector field and optimal transport (Finlay et al., 2020), or even regularization based on the ODE solvers themselves (Pal et al., 2021).

Interestingly, none of these techniques so far make explicit use of the distinct structure and constraints of forecasting problems. In autoregressive models, covariance matrix based regularization has been proposed (Bickel & Gel, 2011). There have been some approaches to the problem using matrix factorization for time series forecasting (Yu et al., 2016; Chen & Sun, 2022). In more general frameworks, temporal attention based methods to guide learning for different time-horizons

can be applied (Fan et al., 2019). Dependence on different prediction horizons also serves to implicitly regularize over the observed intervals (Challu et al., 2022). Hierarchical time series models use predictions at different resolutions and enforce consistency through projections (Rangapuram et al., 2021; 2023). The shadowing lemma (Pilyurin, 1999) has been used to justify estimates of long-term invariants of systems using numerical solvers (Wang et al., 2014; Lasagna et al., 2019). Finally, in systems governed by linear ODEs, it has been observed that the solution space forms a finite-dimensional linear subspace, an observation which can be used to efficiently estimate best fit trajectories and introduce regularization terms based on the Green's functions of the system (González et al., 2014; Mutny & Krause, 2022).

## 1.2 CONTRIBUTIONS

- We propose a principled method for forecasting different time-horizons which extend beyond the duration of the observed data and do not require separate multi-horizon optimization.
- We prove that the space of trajectories of an ODE $\dot{\mathbf{x}} = f(\mathbf{x})$ on any compact interval $I \subset \mathbb{R}$ is a finite-dimensional Riemannian manifold embedded in the space of square integrable functions if $f$ is Lipschitz and continuously differentiable.
- We characterize the transformation from initial conditions and parameters to the trajectory manifold given any Lipschitz and continuously differentiable ODE, thereby enabling optimization on the manifold of feasible ODE trajectories.
- We analyze implications for standard estimation approaches such as maximum likelihood (ML), maximum a posteriori (MAP), and minimum mean squared error (MMSE) estimation, thus enabling the inheritance of their respective statistical guarantees to forecasting.

## 2 PROBLEM FORMULATION

We assume that the underlying data comes from the state space model

$$\dot{\mathbf{x}}_t = f(\mathbf{x}_t, \mathbf{u}_t, \boldsymbol{\theta}, t) \tag{1}$$

$$\mathbf{y}_i \sim \mathcal{P}_{\text{obs}}(\mathbf{x}_{\tau_i}, \boldsymbol{\theta}) \tag{2}$$

$$\mathbf{u} \sim \mathcal{P}_{\text{input}} \tag{3}$$

where $\mathbf{x}_t \in \mathcal{X} \subset \mathbb{R}^N$ is the state of the system at time $t$, $\boldsymbol{\theta} \in \Theta \subset \mathbb{R}^M$ is an unknown set of parameters for the model, $t \in I$ represents time on some finite-interval $I$, $\mathbf{u} \in \mathcal{U} \subset C^1(I, \|\cdot\|_\infty)$ is a continuously differentiable external input, $\mathbf{y}_i$ represents the observation at time $\tau_i$, $\mathcal{P}_{\text{obs}}$ is the observation distribution parameterized by the current state and system parameters, and $\mathcal{P}_{\text{input}}$ is the distribution of system inputs. We let $\mathbf{x}$ and $\mathbf{u}$ represent the entire trajectory and input respectively. In general, the forecasting interval $I$ extends significantly beyond the final measurement time $\tau_i$.

The goal in this work is to enable the use of powerful statistical estimation methods such as maximum likelihood (ML), maximum a posteriori (MAP), and minimum mean squared error (MMSE) estimators to jointly estimate $\mathbf{x}$ over the entire forecasting interval $I$. While these estimators each come with numerous theoretical guarantees, they require the space on which they operate to be well-behaved. We require two assumptions to provide our guarantees of such a structure in this manuscript.

First, the assumption of Lipschitz continuity of the vector field $f$ allows the invocation of the existence and uniqueness theorem, while smoothness implies a smooth dependence on initial conditions and parameters (Khalil, 2002).

**Assumption 1** (Existence, Uniqueness, and Smoothness of Trajectories). *The vector field $f$ is Lipschitz continuous with a continuously differentiable derivative in the forecasting horizon.*

Second, we restrict the space of inputs, initial conditions, and parameterization to be finite dimensional. This enables the usage of tools from differential geometry to transport quantities between manifolds.

**Assumption 2** (Finite-Dimensional Spaces). *The space of possible inputs, $\mathcal{U}$, the state space, $\mathcal{X}$, and the parameter space, $\Theta$, are a finite-dimensional smooth manifolds with or without boundary.*

Under these assumptions, the main contribution of this work is to prove that there exists a smooth isomorphism $\psi : \mathcal{X} \times \mathcal{U} \times \Theta \to \mathcal{C}_{f,I}$, where $\mathcal{C}_{f,I} := \{\mathbf{x} : \dot{\mathbf{x}}_t = f(\mathbf{x}_t, \mathbf{u}_t, \boldsymbol{\theta}, t)\}$ is the space of feasible solutions of (1).

## 3 TRAJECTORY MANIFOLD

The main contribution of this work is the characterization of the finite-dimensional manifold of trajectories of the system in Equation (1), or $\mathcal{C}_{f,I}$. In this section, we introduce a theorem which enables the application of common point estimation techniques to the forecasting problem. In particular, we show that $\mathcal{C}_{f,I}$ is a Riemannian manifold and that $\psi$ represents a smooth transformation onto $\mathcal{C}_{f,I}$. As $\psi$ and its directional derivatives can be readily computed numerically using ODE solvers, this characterization is sufficient for statistical estimation on $\mathcal{C}_{f,I}$. While the full proof is available in Appendix B, we include an outline of the proof here.

**Theorem 1** (Isomorphism Between State Space and Trajectory Space). *Under Assumption 1 and Assumption 2, the space of trajectories $\mathcal{C}_{f,I}$ is a finite-dimensional Riemannian manifold. Furthermore, the transformation $\psi$ defined such that*

$$\psi(\mathbf{x}_0, \mathbf{u}, \boldsymbol{\theta})(t) = \mathbf{x}_0 + \int_0^t f(\mathbf{x}_\tau, \mathbf{u}_\tau, \boldsymbol{\theta}, \tau) d\tau \tag{4}$$

*for all $t \in I$ is a smooth isomorphism between $\mathcal{X} \times \mathcal{U} \times \Theta$ and $\mathcal{C}_{f,I}$.*

*Proof.* Complete proof in Appendix B. □

The proof is completed in three parts, each providing an additional level of structure to $\mathcal{C}_{f,I}$. In each step, we use properties of the flow of the system, or the semigroup of functions $\varphi^\tau : \mathbf{x}_t \mapsto \mathbf{x}_{t+\tau}$ which advance time. We begin by showing that $\psi$ is an injective function into the space of continuous bounded functions on the interval $I$, or $C(I, \|\cdot\|_\infty)$. We then show $\psi$ and $\psi^{-1}$ are continuous to demonstrate the topological manifold structure. Second, we prove that $\mathcal{C}_{f,I}$ is a smooth manifold by showing $\psi$ and $\psi^{-1}$ are continuously differentiable and full rank. Finally, by recalling that $L^p$ spaces on compact subsets of the real line are nested, we inherit the Riemannian metric from $L^2$.

The main consequence of Theorem 1 is that it reduces problems in forecasting to one of propagating a probability distribution through a smooth function. For this reason, the distinctions between the initial conditions, parameters, and inputs are inconsequential, and so for notational clarity we consider $\psi$ to be only a function of the initial condition $\mathbf{x}_0$ for the remainder of this work.

An important note is that the Riemannian metric in $\mathcal{C}_{f,I}$ need not be induced by the standard $L^2$ inner product. A natural extension is to select some symmetric, positive definite integral kernel $K : I \times I \to \mathbb{R}^{N \times N}$ and define the inner product

$$\langle \mathbf{x}, \mathbf{x}' \rangle_K = \int_{I \times I} \mathbf{x}_\tau^\top K(\tau, \tau') \mathbf{x}'_{\tau'} d\tau d\tau', \tag{5}$$

such that $\langle \mathbf{x}, \mathbf{x} \rangle_K > 0$ for any $\mathbf{x} \neq 0$. While the choice of an appropriate integral kernel $K$ may be an interesting independent question, an immediate application is in weighting the importance of different time-horizons. That is, let

$$K(\tau, \tau') = \begin{cases} g(\tau)\mathbf{1} & \tau = \tau' \\ 0 & \text{otherwise} \end{cases}, \tag{6}$$

for some strictly positive $g > 0$ where $\mathbf{1} \in \mathbb{R}^{N \times N}$ is an identity matrix. We thus enable the ability to directly optimize for different forecasting objectives in a statistically rigorous manner **with no major changes to the underlying prediction algorithm**.

## 4 POINT ESTIMATION ON TRAJECTORY MANIFOLDS

In this section, we introduce the statistical tools associated with the trajectory manifold defined in Section 3. In particular, we identify the required fundamental modifications to ML, MAP, and MMSE estimation on the trajectory manifold. In doing so, we additionally specialize the formula for pushing densities along smooth maps between Riemannian manifolds to the transformation between the state-space and the trajectory space.

The generalization of the change of variables formula for random variables to smooth transformations between smooth manifolds is well known. The action is known as a pullback of densities and is

identical to the standard formula, but expressed in local coordinates (Lee, 2013). That is, given some continuous probability density $p_0(\mathbf{x}_0)$ over the initial conditions of the system, $\mathbf{x} = \psi(\mathbf{x}_0)$ is distributed according to

$$p(\mathbf{x}) = |\det D\psi|^{-1} p_0(\psi^{-1}(\mathbf{x})), \tag{7}$$

where $D\psi$ represents the matrix of partial derivatives in local coordinates of $\mathcal{C}_{f,I}$ and $\mathcal{X}$. See Appendix C for additional details.

Local coordinates introduce an inherent difficulty in numerical computations, and so we include the following proposition as an alternative approach to computing $|\det D\psi|$.

**Proposition 1.** *Let $\{\mathbf{v}_i\}$ form an orthonormal basis of the tangent space of $\mathcal{X}$ at $\mathbf{x}_0$, and let $D_{\mathbf{v}_i}|_{\mathbf{x}_0}\psi$ be the directional derivative of $\psi$ in the direction $\mathbf{v}_i$ at $\mathbf{x}_0$, the result of which is represented in the ambient space $L^2(I)$. Finally, define*

$$a_{i,j} := \left\langle D_{\mathbf{v}_i}\big|_{\mathbf{x}_0} \psi, \, D_{\mathbf{v}_j}\big|_{\mathbf{x}_0} \psi \right\rangle_K \quad and \quad A_{\mathbf{x}} := \begin{bmatrix} a_{1,1} & a_{1,2} & \cdots & a_{1,N} \\ a_{2,1} & a_{2,2} & \cdots & a_{2,N} \\ \vdots & \vdots & \ddots & \vdots \\ a_{N,1} & a_{N,2} & \cdots & a_{N,N} \end{bmatrix}, \tag{8}$$

*based on the inner products of the directional derivatives with respect to the basis. Then*

$$|\det D\psi| = \sqrt{|\det A_{\mathbf{x}}|}. \tag{9}$$

*Proof.* Proof available in Appendix D. $\square$

The complexity in this statement is in the correct interpretation of $D\psi$. At first glance, $D\psi$ does not appear to be square, and so the determinant would not be well defined. This potential concern is unfounded though, as $D\psi$ is defined in terms of the tangent spaces of the manifolds, and thus can be represented as a square matrix. A detailed proof is included in Appendix D.

We now connect the geometric properties to common estimation objectives. In particular, we provide methods for MAP estimation, ML estimation, and MMSE estimation constrained to $\mathcal{C}_{f,I}$ or on the ambient space.

## 4.1 ML Estimation

We consider maximum likelihood estimation of the trajectory. Suppose we have some likelihood function of our observations parameterized by the state of the system, e.g. $p(\mathbf{y}_i|\mathbf{x}_{\tau_i})$. Then the likelihood of the entire set of observations can be expressed in terms of the initial condition as $p(\mathbf{y}|\mathbf{x}) = \prod_i p(\mathbf{y}_i|\varphi^{\tau_i}(\mathbf{x}_0))$, and the maximum likelihood estimate of the initial condition is $\arg\max_{\mathbf{x}_0} p(\mathbf{y}|\mathbf{x}_0)$. ML estimation is known to commute with bijective transformations, and thus

$$\hat{\mathbf{x}}_{\mathrm{ML}} = \arg\max_{\hat{\mathbf{x}} \in \mathcal{C}_{f,I}} p(\mathbf{y}|\psi^{-1}(\hat{\mathbf{x}})) = \psi\left(\arg\max_{\hat{\mathbf{x}}_0 \in \mathcal{X}} p(\mathbf{y}|\hat{\mathbf{x}}_0)\right) \tag{10}$$

*Invariant to $\psi$*: It is a well-known that ML estimation commutes with bijective reparameterizations (Trees, 2001). Thus, the ML trajectory in $\mathcal{C}_{f,I}$ is the result of applying $\psi$ to the ML estimate state.

*Invariant to $K$*: The application of the integral kernel $K$ can be viewed as a linear reparameterization. Thus, the ML estimate is invariant to $K$.

## 4.2 MAP Estimation

The behavior of MAP estimation is more subtle than that of ML estimation. While it only involves the addition of a prior to the initial condition, the shift in interpretation from frequentist statistics to Bayesian statistics requires valid probability distributions. Thus, MAP estimation loses invariance to reparameterization and requires the application of Proposition 1. The posterior distribution of the initial condition is $p(\mathbf{x}_0|\mathbf{y}) \propto p(\mathbf{y}|\mathbf{x}_0)p(\mathbf{x}_0)$. To complete MAP estimation on $\mathcal{C}_{f,I}$, we complete a pointwise multiplication of the posterior distribution of the current state with $|\det A_{\mathbf{x}}|^{-1}$, or

$$\hat{\mathbf{x}}_{\mathrm{MAP}} = \arg\max_{\hat{\mathbf{x}} \in \mathcal{C}_{f,I}} p(\hat{\mathbf{x}}|\mathbf{y}) = \arg\max_{\hat{\mathbf{x}} \in \mathcal{C}_{f,I}} p(\psi^{-1}(\hat{\mathbf{x}})|\mathbf{y})p(\psi^{-1}(\hat{\mathbf{x}})) |\det A_{\hat{\mathbf{x}}}|^{-1}, \tag{11}$$

where $\mathbf{x}_0 = \psi^{-1}(\mathbf{x})$ is the initial condition of the trajectory.

*Dependent on $\psi$*: MAP estimation is only invariant to linear reparameterizations (Trees, 2001). This can be seen directly through the dependence on $A_{\mathbf{x}}$ in Equation (11).

*Dependent on $K$*: While a linear transformation to a space ordinarily results in no change to MAP estimation, due to the geodesic curvature of the manifold, the linear transformation becomes nonlinear. This can be seen in Equation (11), which has a non-linear dependence on $\mathbf{x}$ through $K$ as part of $A_{\mathbf{x}}$.

### 4.3 MMSE Estimation on the Ambient Space

MMSE estimation is well-known to be the conditional expectation, or

$$\hat{\mathbf{x}}_{\text{MMSE}} = \arg\min_{\hat{\mathbf{x}} \in L^2(I)} \mathbb{E}\left[\|\hat{\mathbf{x}} - \mathbf{x}\|^2 \mid \mathbf{y}\right] = \mathbb{E}\left[\psi(\mathbf{x}_0) \mid \mathbf{y}\right], \tag{12}$$

where $L^2(I)$ is the space of square integrable functions on $I$.

*Dependent on $\psi$*: Conditional expectation does not in general commute with $\psi$, and so MMSE estimation of the state is a different problem than MMSE estimation of the trajectory.

*Invariant to $K$*: As conditional expectation commutes with linear transformations, the MMSE estimate is invariant to the choice of $K$. An implication is that the MMSE trajectory estimate **is optimal for any desired weighting of time horizons by the construction of Equation** (6). Additional details are provided in Appendix E.

### 4.4 MMSE Estimation on the Manifold

We choose to consider the ambient distance rather than the intrinsic distance on the manifold as the former is both more physically meaningful and computationally tractable. By the orthogonality principle, the MSE of any other estimate $\tilde{\mathbf{x}}$ is the sum of the MSE of this estimate with the squared distance, or $\mathbb{E}\left[\|\tilde{\mathbf{x}} - \mathbf{x}\|^2 \mid \mathbf{y}\right] = \mathbb{E}\left[\|\tilde{\mathbf{x}} - \hat{\mathbf{x}}_{\text{MMSE}}\|^2 \mid \mathbf{y}\right] + \mathbb{E}\left[\|\hat{\mathbf{x}}_{\text{MMSE}} - \mathbf{x}\|^2 \mid \mathbf{y}\right]$. Thus, the MMSE estimate on the manifold is the projection of the ambient MMSE estimate onto the manifold, or

$$\hat{\mathbf{x}}_{\text{MMSE},\mathcal{C}_{f,I}} = \arg\min_{\hat{\mathbf{x}} \in \mathcal{C}_{f,I}} \mathbb{E}\left[\|\hat{\mathbf{x}} - \mathbf{x}\|^2 \mid \mathbf{y}\right] = \arg\min_{\hat{\mathbf{x}} \in \mathcal{C}_{f,I}} \|\hat{\mathbf{x}} - \hat{\mathbf{x}}_{\text{MMSE}}\|^2. \tag{13}$$

*Dependent on $\psi$*: Identical to the ambient case, the MMSE estimate is dependent on $\psi$.

*Dependent on $K$*: $K$ acts in a nonlinear manner on the space through the projection in Equation (13).

## 5 Computation of Estimates

In this section, we describe methods for computing estimates on the trajectory manifold. The core idea is to pull the costs on the manifold into the state space along $\psi$.

**ML Estimation**   ML estimation can be computed in exactly the approach proposed by neural ODEs (Chen et al., 2018). The derivative can be computed using adjoint sensitivity analysis, then standard first-order methods can be applied to best fit the trajectory to the observations. By letting $D\varphi^{\tau_i}$ denote the Jacobian of the flow, the gradient of the log-likelihood is computed in $\mathcal{X}$ as

$$\nabla_{\mathbf{x}_0} \log p(\mathbf{y}|\mathbf{x}) = \sum_i \nabla_{\mathbf{x}_0} \log p(\mathbf{y}_i|\varphi^{\tau_i}(\mathbf{x}_0)) = \sum_i [D\varphi^{\tau_i}] \nabla_{\varphi(\mathbf{x}_\tau)} p(\mathbf{y}_i|\mathbf{x}_\tau). \tag{14}$$

**MAP Estimation**   MAP estimation requires the computation of the reparameterization weighting term which depends on the first derivative of the ODE with respect to initial conditions. Thus, while the derivative exists in principle, it is significantly more expensive to compute numerically through ODE solvers due to dependence on the Hessian of the ODE solution. A full description of the computation of the pushforward weight is available in Appendix F, as well as a discussion of numerical tolerance selection.

For this reason, we propose the usage of zero-order methods to approximate the derivative, or other derivative-free optimization methods such as simulated annealing. Note that this limitation makes MAP estimation significantly less practical than the other techniques as the dimensionality scales.

**MMSE Estimation — Ambient Space**  MMSE Estimation in the ambient space can be computed through a sampling approach. We can construct an approximation of the MMSE estimate as

$$\hat{\mathbf{x}}_{\text{MMSE}} = \mathbb{E}\left[\psi(\mathbf{x}) \mid \mathbf{y}\right] \approx \frac{1}{S}\sum_{i=1}^{S}\psi(\mathbf{X}_{0,i}), \tag{15}$$

where $\{\mathbf{X}_{0,i}\}$ are a set of i.i.d. samples from the posterior distribution of the initial condition, or $p(\mathbf{x}_0|\mathbf{y}) \propto p(\mathbf{y}|\mathbf{x}_0)p(\mathbf{x}_0)$. Often, sampling directly from the posterior is not practical. In such a case, observe that we can readily evaluate the posterior up to a multiplicative scalar. This is sufficient for importance sampling and numerous Markov chain Monte Carlo (MCMC) methods.

**MMSE Estimation — Trajectory Manifold**  MMSE estimation on the manifold can be completed in two steps through the orthogonality principle. First, construct the ambient MMSE estimate. The projection can be computed through gradient-based methods through a geometric pullback of the gradient into the statespace. That is

$$\nabla_{\mathbf{x}_0}\|\hat{\mathbf{x}} - \hat{\mathbf{x}}_{\text{MMSE}}\|^2 = [D\psi]\left(\hat{\mathbf{x}} - \hat{\mathbf{x}}_{\text{MMSE}}\right), \tag{16}$$

where $D\psi$ can be approximated through numerical differentiation through the ODE solver.

## 6 Numerical Experiments

In this section, we include numerical simulations to elucidate the differences in behavior between the different estimation objectives discussed in this work. Throughout our simulations, we compared the performance of the optimal solutions on six different forecasting objectives: ML estimation, MAP estimation of the initial condition, MMSE estimation of the initial condition, MAP estimation of the trajectory, MMSE estimation of the trajectory in the ambient space, and MMSE estimation of the trajectory restricted to $\mathcal{C}_{f,I}$. These objectives are equally distributed between the classical two-step approach of estimating the system state before solving the ODE and direct optimization over the forecasting interval in order to best illustrate the differences in behavior of the solutions.

The key operations in this work are available as a Python library which takes vector fields describing system dynamics as arguments, while the simulation code is included to fully reproduce all figures shown in this section.[1] We implemented our techniques using Diffrax (Kidger, 2021), a library for working with differential equations and machine learning in Jax. Further information on software dependencies, simulation hardware, and simulation details are available in Appendix G.

We simulated the Lotka–Volterra equation,

$$\mathbf{x}_{\tau} = \begin{bmatrix} x_{\tau}^{(1)} \\ x_{\tau}^{(2)} \end{bmatrix} \qquad \dot{\mathbf{x}}_{\tau} = \begin{bmatrix} \alpha x_{\tau}^{(1)} - \beta x_{\tau}^{(1)} x_{\tau}^{(2)} \\ \delta x_{\tau}^{(1)} x_{\tau}^{(2)} - \gamma x_{\tau}^{(2)} \end{bmatrix}, \tag{17}$$

which represents a model of population dynamics between a predator and prey species known to involve oscillations dependent on the initial conditions. This system was chosen in part due to the rapid transitions in the time series, the positions of which are essential in predictions to limit error. We recorded measurements every $0.3$ seconds on the time interval $[0, 3]$ of the form

$$\mathbf{y}_i = \mathbf{x}_{\tau_i} + \eta_i, \tag{18}$$

where $\tau_i$ is the time of the measurement and $\eta_i \sim \mathcal{N}(0, \mathbf{1}\sigma_{\eta}^2)$ is additive i.i.d. Gaussian noise.

We first include a set of experiments to illustrate the behaviors in the different objectives which may lead to poor performance in forecasting tasks. The results of these simulations are shown in Figure 1 and Figure 2: The first includes example trajectories to illustrate issues of oversmoothing and phase mismatch, while the second illustrates the objective function over the state space. Simulations using three additional systems are available in Appendix A.1, demonstrating similar behavior to that in Figure 1. Furthermore, simulations demonstrating the necessity of model knowledge to operate in this data-limited regime are included in Appendix A.2

In Figure 1, the blue and orange lines represent the two different state variables. The dashed lines represent the ground truth, while the solid lines represents the chosen estimate. Qualitatively, we

---

[1]Simulation code available at `https://github.com/{author}/{repository}`.

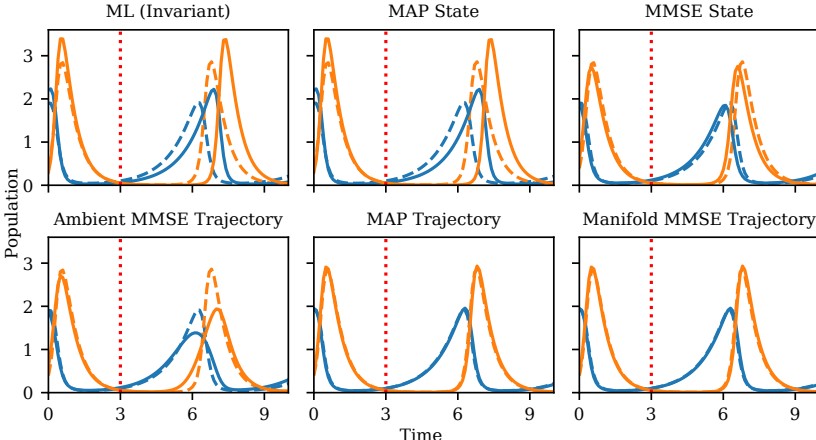

Figure 1: Forecasting Performance based on 6 different objective functions for the Lotka-Volterra equations. The dashed lines indicate the true trajectory of the system, while the solid lines indicate the estimated trajectory. Data collection stopped at the vertical red line.

observe that, despite being the best fit for the observations by construction, the ML and MAP state estimation select trajectories which rapidly lose synchronization with the periodic trajectory. While the MMSE state estimation does better, we see that the second peak is shifted even in this short time horizon. Meanwhile, all three trajectory estimation techniques appear to better match the phase of periodic structure due to the direct dependence in the cost. Finally, while the ambient MMSE trajectory suffers from the commonly seen over-smoothing of forecasts, the manifold constraint maintains the qualitative shape defined by the system at the cost of an increase in MSE.

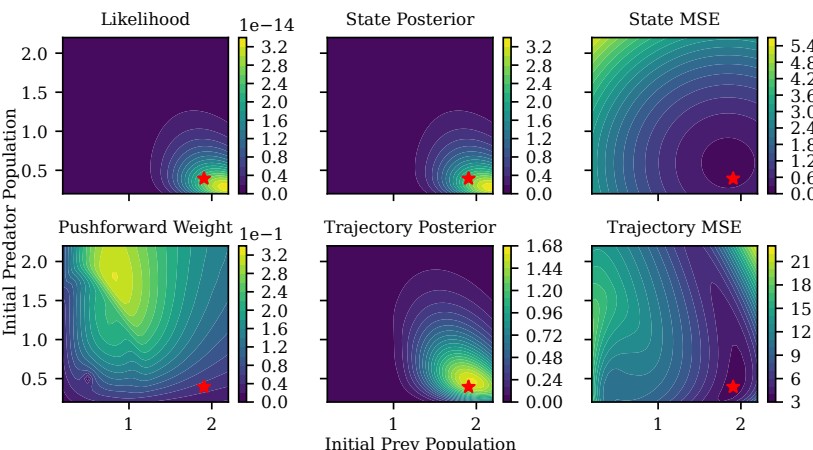

Figure 2: This figure contains examples of the forecasting objectives and pushforward weight in this work for one realization of the Lotka-Volterra system. The red star indicates the true initial condition to be estimated. Each panel contains one objective function, e.g., the likelihood function, the posterior density, or the mean squared error. Notably, the Trajectory MSE plot is the only method to capture the valley of trajectories similar to the true solution.

In Figure 2, we illustrate the objective functions defined by the observations in these simulations, as well as the pushforward weight required to transform between the state space and the trajectory manifold. Observe that the pushforward weight shifts peaks towards regions which are less sensitive to the initial condition. Similarly, the trajectory MSE illustrates a valley of initializations which lead to similar trajectories along the interval, a structure not captured by any competing technique.

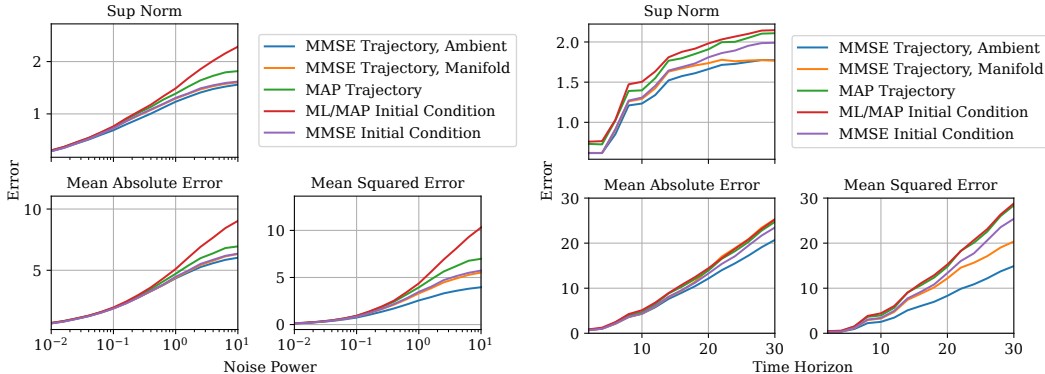

Figure 3: Error as a function of noise power and time horizon for forecasting the Lotke-Volterra system. Left: Forecasting performance as a function of noise power in the observations; Right: Forecasting performance as a function of time horizon.

### 6.1 QUANTITATIVE COMPARISON

In this section, we completed Monte Carlo simulations to compare the MSE, Mean Absolute Error (MAE), and expected sup norm of the error, or $\mathbb{E}\left[\sup_\tau \|\hat{\mathbf{x}}_\tau - \mathbf{x}_\tau\|\right]$ in the trajectory for the proposed objectives as a function of noise power and time horizon. In these simulations, a uniform prior over an interval was chosen, resulting in an identical objective for ML estimation and MAP estimation of the initial condition. Results are shown in Figure 3, where the left panel varies $\sigma_\eta^2$ with a constant 10 second time horizon, while the right panel varies the time horizon with a fixed $\sigma_\eta^2 = 1$.

The key feature in the results is that the maximum likelihood curve, which represents fitting the observed data, always performs the worst, and that this issue becomes even more prominent in longer time horizons and when the noise power is high. This demonstrates the requirement to critically consider the implications of the reparameterization on the time series forecasting problem, particularly when working with time horizons significantly longer than the observation interval.

While the unconstrained MMSE trajectory performs significantly better in MSE than all competing methods, recall that it produces trajectories which do not resemble the original system. While preserving the system structure, MMSE estimation constrained to $\mathcal{C}_{f,I}$ still significantly outperforms the other competing methods, particularly in long time horizons. Furthermore, the performance of the proposed constrained MMSE estimation is often comparable to the performance of the unconstrained solution in MAE and expected sup norm.

## 7 CONCLUSION

In this work, we introduced a method for provably accurate forecasting of time series governed by ODEs through the usage of objectives explicitly dependent on the future trajectory of the system. By proving that the space of finite-horizon trajectories of a continuously differentiable, Lipschitz dynamical system forms a Riemannian manifold, the problem can be described as one of point estimation in a finite-dimensional space. This realization enabled the application of ML, MAP, and MMSE estimation directly in the space of feasible ODE trajectories, where the objectives can be optimized computationally by transporting them into the original state space. Each of these estimators then inherit their respective performance guarantees from the point estimation counterparts: something lacking from the traditional two-step approach of estimating the initial condition before solving the system. The developments in this work will help to provide statistical guarantees on trajectory estimation algorithms, as well as enable the development of new prediction algorithms which include differential equation constraints.

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

# Supplemental Material

## A ADDITIONAL SIMULATIONS

In this section, we include additional simulations in support of this work. In particular, we show simulations for additional systems and comparisons with algorithms which lack the known differential equation structure.

### A.1 ADDITIONAL SIMULATIONS — MMSE

In this section, we include qualitative simulations for three additional dynamical systems: the Van der Pol Oscillator, the Lorenz system, and the Lorenz 96 system. While the Van der Pol Oscillator is a common example of a system exhibiting a limit cycle, the Lorenz and Lorenz 96 systems are examples of chaotic systems. For each of these systems, we plot the true trajectory, the MMSE ambient trajectory, the solution of the when initialized with the MMSE initial conditions, and the proposed manifold-constrained MMSE trajectory. The results of the simulations are shown in Figure 4, where each panel represents a different state variable.

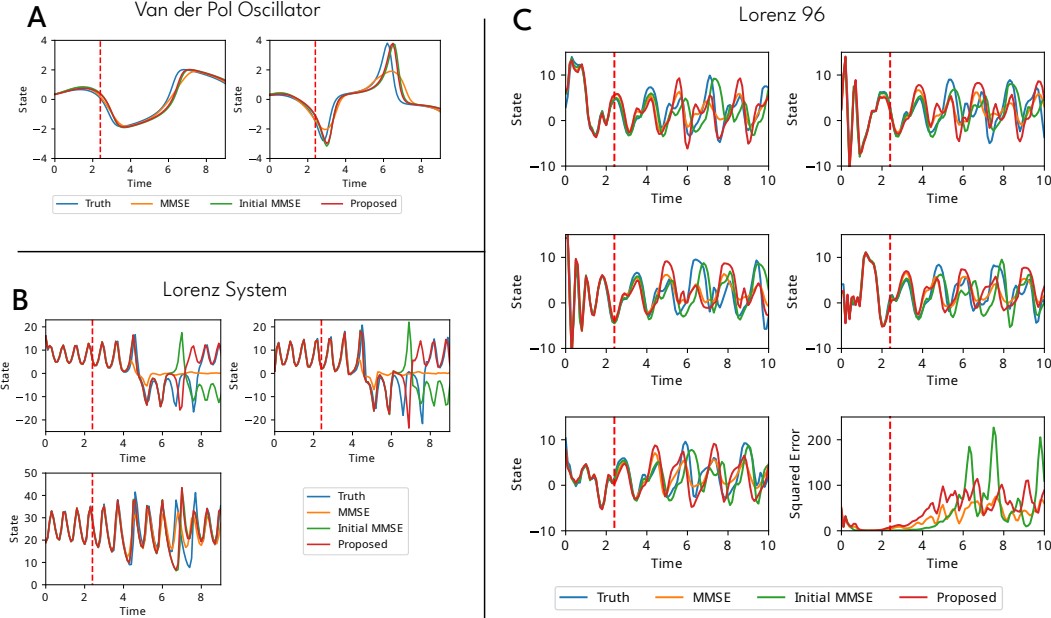

Figure 4: Qualitative evaluation of MMSE forecasting on the trajectory manifold on the Van der Pol Oscillator (A), Lorenz system (B), and the Lorenz 96 system (C) relative to unconstrained MMSE forecasting and MMSE estimation of the initial condition. Similar to Lotka-Volterra simulations in the original manuscript, the unconstrained MMSE forecast suffers from oversmoothing, while the proposed constrained MMSE forecast better preserves the structure. The bottom-right plot of Panel D shows that the proposed method maintains a more consistent level of error over long time horizons than estimation of the initial condition.

The key features to observe are the same as in Figure 1. The MMSE estimate suffers from over-smoothing: This is most clearly visible in Panel B, where the predicted trajectory is nearly constant for two state variables after 5 seconds. Furthermore, we observe that while the initial condition MMSE estimate generally performs well, it suffers from momentary spikes in error which become more prevalent in distant time horizons.

## A.2   IMPORTANCE OF MODEL KNOWLEDGE

In this section, we include a comparison of the manifold-constrained MMSE forecast with Gaussian process regression (GP) and a vector autoregressive models (VAR) to demonstrate the necessity of the known model structure in this data-limited regime. Each model was given 5 samples spaced by 0.6 seconds under additive Gaussian noise. GP used the exp-sine-squared kernel with a white noise kernel to account for measurement noise. The exp-sine-squared kernel was chosen to enable GP to capture the known periodic behavior of the Lotka-Volterra system. We use two versions of GP, one without a known period for the trajectory and one with a known period. The results of the simulations can be seen in Figure 5.

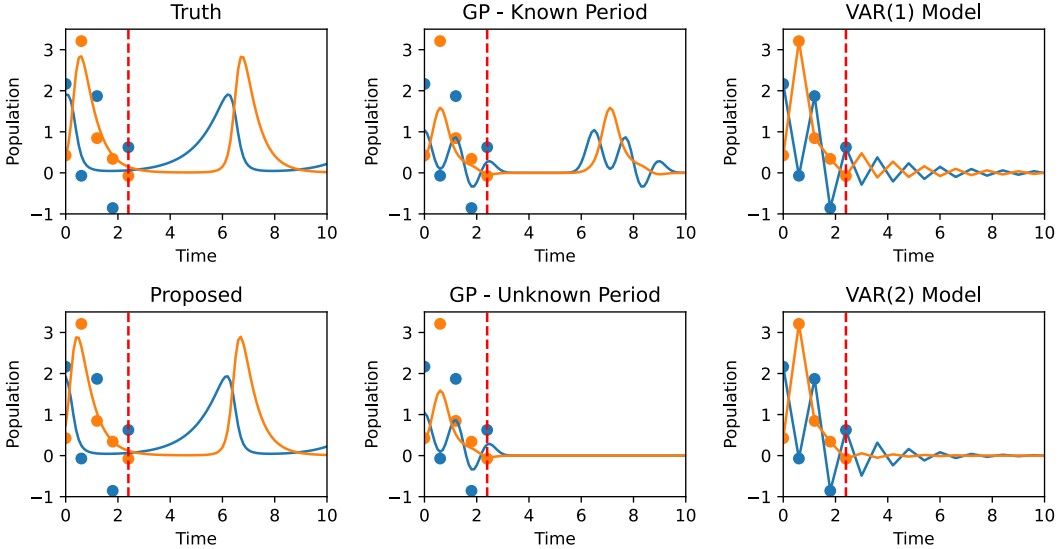

Figure 5: A qualitative comparison of MMSE forecasting in the trajectory manifold, Gaussian process regression (GP) with the exp-sine-squared kernel, and a vector autoregressive process (VAR) using 1 and 2 time lags. Without the added knowledge of the differential equation structure, both GP and VAR fail to extrapolate well. When explicitly given the periodicity, GP still fails to capture the structure of the signal.

In the simulations, we see that the methods without the periodic structure explicitly encoded in the model assumptions are unable to extrapolate correctly based on the limited data. Additionally, the GP method with a known period still fails to capture the general structure of the signal, unlike the proposed manifold-constrained MMSE method. Finally, it is worth noting that the proposed method lacks the apriori knowledge of the period of the signal that was provided to GP. Despite this, the differential equation structure is sufficient to well-approximate it.

## B PROOF OF THEOREM 1

In this section, we present the full proof of Theorem 1, as well as the intermediate results. We make our notation more concise by observing that $\mathbf{u}$ and $\boldsymbol{\theta}$ can be folded into the $\mathbf{x}_0$ for the following reason.

First, note that we can incorporate the input space into the parameter space by considering an augmented parameter space $\Theta' = \Theta \times \mathcal{U}$ to be the product manifold of the two spaces. By Assumption 2, the augmented parameter space is still finite-dimensional. Finally, observe that $\boldsymbol{\theta}$ can be incorporated into $\mathbf{x}$ under the dynamics $\dot{\boldsymbol{\theta}} = 0$. Thus, without loss of generality, we will consider the dynamical system $\dot{\mathbf{x}} = f(\mathbf{x}, t)$ for the remainder of the argument.

We begin by showing that $\psi$ is an injective function into the desired space.

**Lemma 1.** *Let $f$ be Lipschitz in both $\mathbf{x}$ and $t$, then $\psi$ is an injective function into the space of continuous bounded functions, or $C(I, \|\cdot\|_\infty)$.*

*Proof.* For all $\mathbf{x}_0 \in \mathcal{X}$, $\psi(\mathbf{x}_0)$ is a continuous function on a compact interval $\psi(\mathbf{x}_0) \in C(I)$. Continuous functions on compact intervals are bounded, or $\|\psi(\mathbf{x}_0)\|_\infty < \infty$. Thus $\psi(\mathbf{x}_0) \in C(I, \|\cdot\|_\infty)$.

Assume there exists $\mathbf{x}_0, \tilde{\mathbf{x}}_0 \in \mathcal{X}$ such that $\mathbf{x}_0 \neq \tilde{\mathbf{x}}_0$ and $\psi(\mathbf{x}_0) = \psi(\tilde{\mathbf{x}}_0)$. But, by the existence and uniqueness theorem (Khalil, 2002), $\psi(\mathbf{x}_0)(\tau) \neq \psi(\tilde{\mathbf{x}}_0)(\tau)$ for all $\tau$. Thus $\psi(\mathbf{x}_0) \neq \psi(\tilde{\mathbf{x}}_0)$ on the entire interval, and we have a contradiction. $\qquad\square$

We next show that $\psi$ is a homeomorphism by directly showing continuity of both the original function and the inverse. We do so by considering the flow $\varphi : \mathbf{x}_0 \times t \mapsto \mathbf{x}_t$, which is well-known to be continuously differentiable under Assumption 1.

**Lemma 2.** *$\psi$ is continuous with a continuous inverse on its image.*

*Proof. Continuity of $\psi$:*
By continuity of $\varphi$, there exists $\delta(\tau, \epsilon)$ such that $\|\mathbf{x}_0 - \tilde{\mathbf{x}}_0\| < \delta(\tau, \epsilon) \implies |\varphi(\mathbf{x}_0, \tau) - \varphi(\tilde{\mathbf{x}}_0, \tau)| < \epsilon$.

On compact subsets of metric spaces, continuity of a function is equivalent to uniform continuity, and thus the time dependence can be removed. That is

$$\|\mathbf{x}_0 - \tilde{\mathbf{x}}_0\| < \delta(\epsilon) \implies \forall \tau, \ \|\varphi(\mathbf{x}_0, \tau) - \varphi(\tilde{\mathbf{x}}_0, \tau)\| < \epsilon \tag{19}$$

Choose $\mathbf{x}_0, \tilde{\mathbf{x}}_0 \in \mathcal{X}$ such that

$$\|\mathbf{x}_0 - \tilde{\mathbf{x}}_0\| < \delta(\epsilon). \tag{20}$$

We bound the distance between the image of the two points after $\psi$ by first expanding the sup norm in terms of the flow, i.e.

$$\|\psi(\mathbf{x}_0) - \psi(\tilde{\mathbf{x}}_0)\|_\infty = \sup_{\tau \in I} \|\varphi(\mathbf{x}_0, \tau) - \varphi(\tilde{\mathbf{x}}_0, \tau)\| \tag{21}$$

Finally, note that the bound in Equation (19) applies to Equation (21) and thus

$$\|\mathbf{x}_0 - \tilde{\mathbf{x}}_0\| < \delta(\epsilon) \implies \|\psi(\mathbf{x}_0) - \psi(\tilde{\mathbf{x}}_0)\|_\infty \leq \epsilon \tag{22}$$

*Continuity of $\psi^{-1}$:*
Choose $\mathbf{x}, \tilde{\mathbf{x}} \in \mathcal{C}_{f,I}$ such that

$$\|\mathbf{x} - \tilde{\mathbf{x}}\|_\infty < \delta. \tag{23}$$

Expand the sup norm in terms of points of the function, i.e.

$$\|\mathbf{x} - \tilde{\mathbf{x}}\|_\infty = \sup_\tau \|\mathbf{x}_\tau - \tilde{\mathbf{x}}_\tau\| < \delta. \tag{24}$$

Finally, note that $\psi^{-1}(\mathbf{x}) = \mathbf{x}_0$ and that, by definition of the supremum,

$$\|\mathbf{x}_0 - \tilde{\mathbf{x}}_0\| \leq \sup_\tau \|\mathbf{x}_\tau - \tilde{\mathbf{x}}_\tau\| < \delta. \tag{25}$$

Thus $\|\mathbf{x} - \tilde{\mathbf{x}}\|_\infty < \delta(\epsilon) \implies \|\psi^{-1}(\mathbf{x}) - \psi^{-1}(\tilde{\mathbf{x}})\| < \epsilon$, where $\delta(\epsilon) = \epsilon$.

$\qquad\square$

**Corollary 1.** *Let $f$ be Lipschitz and let $\mathcal{X}$ be a topological manifold. Then $\mathcal{C}_{f,I}$ is a topological manifold.*

*Proof.* By Lemma 2, $\psi$ is a homeomorphism onto its image. Thus, it is an embedding. $\qquad\square$

We now proceed to introduce the smooth structure of $\mathcal{C}_{f,I}$.

**Lemma 3.** *Let $f$ be continuously differentiable, then $\psi$ is continuously differentiable.*

*Proof.* The key realization in this proof is that by Corollary 1, $\psi$ is a function between finite-dimensional spaces. Thus, we can prove differentiability based on partial derivatives rather than use an infinite-dimensional framework.

First, note that the flow $\varphi(\mathbf{x}_0, \tau)$ is continuously differentiable with respect to $\mathbf{x}_0$ for each $\tau$. We show that these partial derivatives pointwise in time define a partial derivative over all time.

Let $\mathbf{v}_\tau^{(\mathbf{x}_0)} = \frac{d\varphi(\mathbf{x}_0+\gamma\mathbf{h},\tau)}{d\gamma}$ be the directional derivative for time $\tau$, and consider $\mathbf{v}^{(\mathbf{x}_0)}$ to be the time-dependent function defined by the concatenation of the directional derivatives. Then

$$\left\| \frac{\psi(\mathbf{x}_0 + \gamma\mathbf{h}) - \psi(\mathbf{x}_0)}{\gamma} - \mathbf{v}^{(\mathbf{x}_0)} \right\|_\infty = \sup_\tau \left\| \frac{\varphi(\mathbf{x}_0 + \gamma\mathbf{h}, \tau) - \varphi(\mathbf{x}_0, \tau)}{\gamma} - \mathbf{v}_\tau^{(\mathbf{x}_0)} \right\|. \tag{26}$$

By the mean value theorem, there exists $0 < \gamma'_\tau < \gamma$

$$\frac{\varphi(\mathbf{x}_0 + \gamma\mathbf{h}, \tau) - \varphi(\mathbf{x}_0, \tau)}{\gamma} = \mathbf{v}_\tau^{(\mathbf{x}_0+\gamma'_\tau\mathbf{h})}. \tag{27}$$

By continuity of the derivative of $\varphi$, there exists $\delta(\epsilon, \tau)$ such that

$$\gamma'_\tau < \delta(\epsilon, \tau) \implies \|\mathbf{v}_\tau^{(\mathbf{x}_0+\gamma'\mathbf{h})} - \mathbf{v}_\tau^{(\mathbf{x}_0)}\| < \epsilon. \tag{28}$$

Recall $\gamma'_\tau < \gamma$, and thus

$$\gamma < \delta(\epsilon, \tau) \implies \|\mathbf{v}_\tau^{(\mathbf{x}_0+\gamma'\mathbf{h})} - \mathbf{v}_\tau^{(\mathbf{x}_0)}\| < \epsilon. \tag{29}$$

Finally, observe that compactness in time strengthens continuity to uniform continuity, and thus

$$\gamma < \delta(\epsilon) \implies \sup_\tau \|\mathbf{v}_\tau^{(\mathbf{x}_0+\gamma'\mathbf{h})} - \mathbf{v}_\tau^{(\mathbf{x}_0)}\| < \epsilon. \tag{30}$$

Thus, Equation (26) is upper bounded by $\epsilon$, and $\psi$ is differentiable. The derivative inherits continuity from the pointwise derivative similarly. $\qquad\square$

**Lemma 4.** *Let $f$ be continuously differentiable. Then $\psi^{-1}$ is continuously differentiable.*

*Proof.* $\psi^{-1}$ can be represented by the linear functional $h(f) = \langle \delta_0, f \rangle$, where $\delta_0$ is the Dirac delta function. Thus, as the extension of $\psi^{-1}$ to the ambient space is linear, the extension is differentiable. The derivative of $\psi^{-1}$ is then the projection of the extension onto the tangent space. $\qquad\square$

**Lemma 5.** *Assume that $f$ is Lipschitz and continuously differentiable. Then $\psi$ is a diffeomorphism onto its image.*

*Proof.* Follows directly from Lemma 3 and Lemma 4. The full-rank requirement is satisfied due to the inclusion of $\mathbf{x}_0$ in $\psi(\mathbf{x}_0)$ itself. Thus, initial condition perturbations necessarily result in distinct derivatives. $\qquad\square$

**Theorem 1** (Isomorphism Between State Space and Trajectory Space). *Under Assumption 1 and Assumption 2, the space of trajectories $\mathcal{C}_{f,I}$ is a finite-dimensional Riemannian manifold. Furthermore, the transformation $\psi$ defined such that*

$$\psi(\mathbf{x}_0, \mathbf{u}, \boldsymbol{\theta})(t) = \mathbf{x}_0 + \int_0^t f(\mathbf{x}_\tau, \mathbf{u}_\tau, \boldsymbol{\theta}, \tau)d\tau \tag{4}$$

*for all $t \in I$ is a smooth isomorphism between $\mathcal{X} \times \mathcal{U} \times \Theta$ and $\mathcal{C}_{f,I}$.*

*Proof.* First, note that $\psi(\mathcal{X})$ is a topological manifold by Corollary 1. By Lemma 5, $\mathcal{X}$ and $\psi(\mathcal{X})$ are diffeomorphic. Thus, as $\mathcal{X}$ is a smooth manifold, so is $\psi(\mathcal{X})$. $\qquad\square$

## C   PROBABILITY ON RIEMANNIAN MANIFOLDS

We include here a brief overview of the key definitions in probability on Riemannian manifolds. First, recall the definition of a probability space as the triplet of $(\Omega, \mathcal{F}, P)$, where $\Omega$ is the underlying space, $\mathcal{F}$ is a $\sigma$-algebra on $\Omega$, and $P$ is a probability measure on $\mathcal{F}$ satisfying non-negativity, countable additivity, and $P(\Omega) = 1$. In this section, we will consider the more restrictive case where $P$ can be represented by a probability density function, e.g. $P(B) = \int_B p(\mathbf{x})d\mathbf{x}$.

A related, but distinct notion is a density on a manifold. The following description integration of densities on smooth manifolds is adapted from the textbook *Introduction to Smooth Manifolds* (Lee, 2013).

A density on a manifold is (loosely) a function mapping collections of vector fields into real-valued functions satisfying the following pullback property:

Let $F : \mathcal{M} \to \mathcal{N}$ be a smooth function between two manifolds and let $\mu$ be a density on $\mathcal{N}$. Then $F^*\mu$ is a density on $\mathcal{M}$ given by

$$(F^*\mu)_p(\mathbf{v}_1, \dots \mathbf{v}_N) = \mu_{F(p)}(dF_p(\mathbf{v}_1), \dots, dF_p(\mathbf{v}_N)), \tag{31}$$

where $\{\mathbf{v}_i\}_{i=1}^N$ is a collection of vectors in the tangent space of $\mathcal{N}$ at $p$, and $dF_p$ is the differential of $F$ at $p$. In local coordinates, this becomes

$$F^*(u|d\mathbf{w}^1 \wedge \cdots \wedge d\mathbf{w}^N|) = (u \circ F)|\det DF||d\mathbf{v}^1 \wedge \dots \wedge d\mathbf{v}^N|, \tag{32}$$

where $u$ is a continuous function, $\{\mathbf{w}^i\}$ and $\{\mathbf{v}^i\}$ represent local coordinates, and $\wedge$ represents the wedge product.

Integration is commonly extended to manifolds based on the usage of charts between the manifold and Euclidean space. That is, let $\mathcal{M}$ be a smooth manifold, and let $U \subseteq \mathcal{M}$ be a subset such that it is entirely covered by the domain of the chart $\phi : U \to \mathbb{R}^N$, then

$$\int_U \mu = \int_{\phi(U)} \left(\phi^{-1}\right)^* \mu, \tag{33}$$

where $\left(\phi^{-1}\right)^*$ is the pullback along $\phi$ and $\mu$ is a density on the manifold. The invariance in Equation (33) is in fact a special case of a more general property. That is

$$\int_{\mathcal{M}} \mu = \int_{\mathcal{N}} F^*\mu. \tag{34}$$

Finally, observe that by combining Equation (34) and Equation (32), and taking $p(\mathbf{x})$ to be a density on a manifold, we arrive at the standard probability density reparameterization but in local coordinates, as shown in Equation (7).

## D   PROOF OF PROPOSITION 1

**Proposition 1.** *Let $\{\mathbf{v}_i\}$ form an orthonormal basis of the tangent space of $\mathcal{X}$ at $\mathbf{x}_0$, and let $D_{\mathbf{v}_i}|_{\mathbf{x}_0}\psi$ be the directional derivative of $\psi$ in the direction $\mathbf{v}_i$ at $\mathbf{x}_0$, the result of which is represented in the ambient space $L^2(I)$. Finally, define*

$$a_{i,j} := \left\langle D_{\mathbf{v}_i}|_{\mathbf{x}_0} \psi, \, D_{\mathbf{v}_j}|_{\mathbf{x}_0} \psi \right\rangle_K \quad and \quad A_{\mathbf{x}} := \begin{bmatrix} a_{1,1} & a_{1,2} & \cdots & a_{1,N} \\ a_{2,1} & a_{2,2} & \cdots & a_{2,N} \\ \vdots & \vdots & \ddots & \vdots \\ a_{N,1} & a_{N,2} & \cdots & a_{N,N} \end{bmatrix}, \tag{8}$$

*based on the inner products of the directional derivatives with respect to the basis. Then*

$$|\det D\psi| = \sqrt{|\det A_{\mathbf{x}}|}. \tag{9}$$

This proof comes from considering a unitary transformation from the ambient parameterization of the tangent space of the manifold onto $\mathbb{R}^N$. It explicitly constructs a square matrix representation of $D\psi$, before noting that the unitary transformation vanishes in the matrix product.

*Proof.* Let $U$ be a unitary transformation from the basis of the tangent space of $\mathcal{C}_{f,I}$ at $\psi(\mathbf{x}_0)$ in the ambient space $L^2(I)$ onto the tangent space of $\mathcal{X}$. $U$ can be explicity represented as

$$U\mathbf{x} = \sum_{i=1}^{N} \mathbf{w}_i' \langle \mathbf{w}_i, \mathbf{x} \rangle, \tag{35}$$

where $\{\mathbf{w}_i\}_{i=1}^{N}$ form an orthonormal basis of the tangent space in ambient coordinates and $\{\mathbf{w}_i'\}_{i=1}^{N}$ form an orthonormal basis of the tangent space of $\mathcal{X}$. The adjoint of $U$ can be represented as

$$U^*\mathbf{x}' = \sum_{i=1}^{N} \mathbf{w}_i \langle \mathbf{w}_i', \mathbf{x}' \rangle, \tag{36}$$

and $U^*U$ is clearly an identity. Finally, consider $D\psi$ as a linear operator from $\mathcal{X}$ onto $L^2(I)$, and the action can be expressed through an orthogonal decomposition as

$$(D\psi)(\mathbf{h}_0) = \sum_{i=1}^{N} (D_{\mathbf{v}_i}|_{\mathbf{x}_0}\psi) \langle \mathbf{v}_i, \mathbf{h}_0 \rangle. \tag{37}$$

Then, by linearity, $UD\psi$ represents the Jacobian of the transformation from initial conditions to a Euclidean representation of the tangent space of the manifold of trajectories at $\psi(\mathbf{x}_0)$ and can be readily represented as a square matrix. By multiplicity of determinants,

$$\det\{UD\psi\} = \det\{U\}\det\{D\psi\} = \det\{D\psi\}. \tag{38}$$

Then, through standard properties of determinants, we can expand the original determinant through algebraic manipulations of $|\det\{D\psi\}|^2$ to include $U$, or

$$|\det\{D\psi\}|^2 = |\det\{UD\psi\}|^2 = \left|\det\left\{(UD\psi)^* UD\psi\right\}\right|. \tag{39}$$

We can now arrive at the final statement by expanding the expression.

$$(UD\psi)(\mathbf{h}_0) = \sum_{i=1}^{N} \mathbf{w}_i' \langle \mathbf{w}_i, \sum_{j=1}^{N} (D_{\mathbf{v}_j}|_{\mathbf{x}_0}\psi) \langle \mathbf{v}_j, \mathbf{h}_0 \rangle \rangle \tag{40}$$

$$= \sum_{i=1}^{N} \mathbf{w}_i' \sum_{j=1}^{N} \langle \mathbf{w}_i, D_{\mathbf{v}_j}|_{\mathbf{x}_0}\psi \rangle \langle \mathbf{v}_j, \mathbf{h}_0 \rangle \tag{41}$$

Then by interpreting Equation (41) as a matrix, the adjoint can be recognized as

$$(UD\psi)^*(\mathbf{h}_0') = \sum_{j=1}^{N} \mathbf{v}_j \sum_{i=1}^{N} \langle \mathbf{w}_i, D_{\mathbf{v}_j}|_{\mathbf{x}_0}\psi \rangle \langle \mathbf{w}_i', \mathbf{h}_0' \rangle. \tag{42}$$

Thus,

$$(UD\psi)^*(UD\psi)(\mathbf{h}_0) = \sum_{j=1}^{N} \mathbf{v}_j \sum_{i=1}^{N} \langle \mathbf{w}_i, D_{\mathbf{v}_j}|_{\mathbf{x}_0}\psi \rangle \sum_{j'=1}^{N} \langle \mathbf{w}_i, D_{\mathbf{v}_{j'}}|_{\mathbf{x}_0}\psi \rangle \langle \mathbf{v}_{j'}, \mathbf{h}_0 \rangle \tag{43}$$

$$= \sum_{j=1}^{N} \mathbf{v}_j \sum_{j'=1}^{N} \langle \mathbf{v}_{j'}, \mathbf{h}_0 \rangle \sum_{i=1}^{N} \langle \mathbf{w}_i, D_{\mathbf{v}_j}|_{\mathbf{x}_0}\psi \rangle \langle \mathbf{w}_i, D_{\mathbf{v}_{j'}}|_{\mathbf{x}_0}\psi \rangle \tag{44}$$

$$= \sum_{j=1}^{N} \mathbf{v}_j \sum_{j'=1}^{N} \langle \mathbf{v}_{j'}, \mathbf{h}_0 \rangle \langle D_{\mathbf{v}_j}|_{\mathbf{x}_0}\psi, D_{\mathbf{v}_{j'}}|_{\mathbf{x}_0}\psi \rangle \tag{45}$$

or

$$(UD\psi)^*(UD\psi) = A_{\mathbf{x}}. \tag{46}$$

Thus, by Equation (39),

$$|\det D\psi| = \sqrt{|\det A_{\mathbf{x}}|}. \tag{47}$$

$\square$

## E  INVARIANCE TO $K$ IN SECTION 4.3

In this section, we provide supplemental notes on the invariance of MMSE estimation in the ambient space to the choice of positive-definite integral kernel $K$. First, expand the norm as an $L^2$ inner product, or $\|\mathbf{x}\|_K = \langle K\mathbf{x}, \mathbf{x}\rangle_2$. By the positive definite assumption, let $K = B^*B$, where $B$ is the square root operator. Thus $\|\mathbf{x}\|_K = \|B\mathbf{x}\|_2$. Let $\mathbf{x}' = B\mathbf{x}$, then by optimality of conditional expectation,

$$\hat{\mathbf{x}}'_{\text{MMSE}} = \mathbb{E}\left[\mathbf{x}' \,|\, \mathbf{y}\right] = \mathbb{E}\left[B\mathbf{x} \,|\, \mathbf{y}\right] = B\mathbb{E}\left[\mathbf{x} \,|\, \mathbf{y}\right] = B\hat{\mathbf{x}}_{\text{MMSE}}. \tag{48}$$

Thus, $\arg\min_{\hat{\mathbf{x}}} \mathbb{E}\left[\|\hat{\mathbf{x}} - \mathbf{x}\|_K \,|\, \mathbf{y}\right] = \arg\min_{\hat{\mathbf{x}}} \mathbb{E}\left[\|\hat{\mathbf{x}} - \mathbf{x}\|_2 \,|\, \mathbf{y}\right]$. As $K$ can be selected to apply weighted penalties dependent on the time horizon, the MMSE estimate in the ambient space is optimal for all such weightings.

## F  COMPUTATION OF EQUATION (7)

In this section, we describe the computations required for the application of Proposition 1.

A key observation is that each $a_{i,j}$ in Proposition 1 is readily computable through classical sensitivity analysis tools. Furthermore, the rich history of scientific computing enables guarantees on the accuracy of the computation of each $a_{i,j}$, which can then be related to the accuracy of $\sqrt{|\det DA|}$.

The general approach is to numerically evaluate the sensitivity at a set of grid points in time, $\{\tau_i\}$, then use these grid points to numerically approximate the integral. This approach is described in Algorithm 1. Tolerances and step sizes for the numerical ODE solver can be chosen to guarantee some error bound on the determinant, the design of which is described in Appendix F.1, below. Finally, it is additionally worth noting that $|\det D\psi|$ can be precomputed for a given system, enabling computational efficiency when the data is actually acquired if the ODE is known ahead of time.

---

**Algorithm 1:** Compute $|\det D\psi|$ for a given ODE $\dot{x} = f(x)$ and compact domain $\Omega \subset \mathcal{X}$

**Input:** $f : \mathcal{X} \to \mathcal{X}$, Interval $I = [0, T]$, Initial Condition $x_0$, solver_tolerance, step_size
**Result:** A value $s$ such that $|s - |\det D\psi|| <$ tolerance set by solver_tolerance and step_size
**for** $i = 1 \ldots N$ **do**
  | $U_{i,:}$ = automatic_differentiation(odesolve($f$, $x_0$, step_size, solver_tolerance))
**end**
Construct $A$ through trapezoidal integration over $U$ inner products;
**return** $\sqrt{|\det A|}$

---

### F.1  TOLERANCE SELECTION FOR ALGORITHM 1

In this section, we describe the key terms for selecting the tolerance and step sizes in Algorithm 1. To do so, we chain together bounds on the sensitivity. Because existing methods allow the selection of tolerances for numerical integration, we begin by relating error in $\det A$ to that of its elements $a_{i,j}$.

There exist numerous results on this question, but we select the following pair of bounds on the perturbation of the determinant from (Ipsen & Rehman, 2008)

$$|\det(A + E) - \det(A)| \leq N\|E\|_2 \max\left\{\|A\|_2, \|A + E\|_2\right\}^{N-1} \qquad \text{(Absolute)} \tag{49}$$

$$\frac{|\det(A + E) - \det(A)|}{|\det(A)|} \leq \left(\kappa \frac{\|E\|_2}{\|A\|_2} + 1\right)^N - 1, \qquad \text{(Relative)} \tag{50}$$

where $\kappa$ is the condition number of $A$. Thus, we can turn a constraint in the determinant into a constraint on the induced 2-norm of the error matrix.

We can then use standard norm equivalences to note that

$$\|E\|_2 \leq N\|E\|_{\max}, \tag{51}$$

where $E \in \mathbb{R}^{N \times N}$ and $\|\cdot\|_{\max}$ denotes the maximum value in the matrix.

As each element of $E$, or $e_{i,j}$, represents the error in the computation of $a_{i,j}$, we now consider the error in the computation of the integral. If we denote the true directional derivatives to be functions $u_i : I \to \mathcal{X}$, then

$$a_{i,j} = \int_I \langle u_i(t), u_j(t) \rangle dt. \tag{52}$$

We approximate each $a_{i,j}$ integral as a summation using trapezoidal rule and thus

$$|\tilde{e}_{i,j}| \leq \frac{T^3}{12n^2} \max_{\tau \in I} \left\{ \left. \frac{\partial^2}{\partial t^2} \langle u_i(t), u_j(t) \rangle \right|_{t=\tau} \right\}, \tag{53}$$

where $\tilde{e}_{i,j}$ is the error under exact computation of the derivative and $n$ is the number of steps. Finally, note that we do not have samples of the exact derivative, but instead values perturbed by at most $\text{tol}_{ode}$, or the tolerance of the numerical computation of the derivative. Thus, we must additionally include the square of this error, or

$$|e_{i,j}| \leq \frac{T^3}{12n^2} \max_{\tau \in I} \left\{ \left. \frac{\partial^2}{\partial t^2} \langle u_i(t), u_j(t) \rangle \right|_{t=\tau} \right\} + \text{tol}_{ode}^2 T \tag{54}$$

Assembling the inequalities, the relative tolerance is upper bounded as

$$\frac{|\det(A+E) - \det(A)|}{|\det(A)|} \leq \left( \kappa \frac{\|E\|_2}{\|A\|_2} + 1 \right)^N - 1 \tag{55}$$

$$\leq \left( N\|A^{-1}\|_2 \|E\|_{\max} + 1 \right)^N - 1 \tag{56}$$

$$\leq \left( N\|A^{-1}\|_2 \left( \frac{T^3}{12n^2} \max_{\tau \in I} \left\{ \left. \frac{\partial^2}{\partial t^2} \langle u_i(t), u_j(t) \rangle \right|_{t=\tau} \right\} + T\text{tol}_{ode}^2 \right) + 1 \right)^N - 1. \tag{57}$$

Finally, to choose the grid resolution and ODE solver tolerance, we require bounds on $\|A^{-1}\|_2$ and $\max_{\tau \in I} \left\{ \left. \frac{\partial^2}{\partial t^2} \langle u_i(t), u_j(t) \rangle \right|_{t=\tau} \right\}$.

For $\|A^{-1}\|$, we can use a Gersgorin-type lower bound on $A$ such as (Johnson, 1989) to see

$$\|A^{-1}\|_2 \leq \left( \min_i \left\{ |a_{i,i}| - \sum_{j \neq i} |a_{i,j}| \right\} \right)^{-1}. \tag{58}$$

Finally, the absolute tolerance can be similarly bounded as

$$|\det(A+E) - \det(A)| \leq N\|E\|_2 \max \left\{ \|A\|_2, \|A+E\|_2 \right\}^{N-1} \tag{59}$$

$$\leq N\|E\|_{\max} \left( \|A\|_2 + \|E\|_{\max} \right)^{N-1}, \tag{60}$$

where $\|E\|_{max}$ is dependent on the inverse square of the step size and the square of the ODE solver tolerance.

From these results, we can see a key property. First, the relative tolerance is bounded by a polynomial in $n^{-1}$ and $\text{tol}_{ode}$. If $n^{-1} = \text{tol}_{ode}$, then the convergence becomes quadratic in the tolerance level. Thus, this property gives guidance on the trade-off between computation time and accuracy to push densities onto $\mathcal{C}_{f,I}$.

# G SIMULATION DETAILS

All simulations were completed using a regularly sampled grid in order to increase the throughput for the Monte Carlo simulations through precomputed transformations. The simulations can be precomputed due to the linearity of the transformations in the space of probability densities. Thus, a grid-based approach allows the reduction of the inference steps to an elementwise multiplication of tensors and a selection of the maximum or minimum value. While this approach is slower when computing a single estimate due to the initial overhead, it significantly accelerates the process for low-dimensional problems in successive applications.

While the full simulation code is available in the git repository with the library, Table 1 contains the key parameters in the simulations. As the error is dominated by that of the statistical inference, fast, but low accuracy, solvers were used for the ODE.

Table 1: Simulation Details

| | |
|---|---|
| Feasible Initial Conditions | $[0.2, 2.2] \times [0.2, 2.2]$ |
| Grid Resolution | $0.05$ |
| Monte Carlo Samples | $10,000$ |
| Lotke-Volterra Parameters | $\alpha = 1;\ \beta = 2;\ \delta = 4;\ \gamma = 2$ |
| ODE Solver Algorithm | Heun's Method |
| ODE Solver Tolerance | rtol: $10^{-2}$; atol: $10^{-2}$ |
| ODE Saved Timestep Step Size | $0.1$ |

## G.1 LIBRARY DEPENDENCIES

The library developed for this work requires the dependencies listed in Table 2.

Documentation was generated using Sphinx, which uses the BSD license.

Testing is done through Pytest 7.3.1, which uses the MIT license.

Table 2: Library Dependencies

| Dependency | Version | License |
|---|---|---|
| Python | $\geq 3.9$ | PSF |
| Jax | $\geq 0.4.3$ | Apache-2.0 |
| Diffrax | $\geq 0.3.0$ | Apache-2.0 |
| Jaxtyping | $\geq 2.0.0$ | MIT |

## G.2 PLOTTING AND SIMULATION DEPENDENCIES

Additional library versions at time of simulation are shown in Table 3.

Table 3: Further Simulation Dependencies

| Dependency | Version | License |
|---|---|---|
| Matplotlib | 3.7.1 | BSD-compatible |
| Numpy | 1.24.2 | BSD |
| TQDM | 4.65.0 | Mix of MIT and MPL |

## G.3 SIMULATION HARDWARE

Simulations were run on a shared server without job scheduling.

The server hardware is available in Table 4.

Table 4: Server Hardware

| Category | Component |
|----------|-----------|
| CPU | AMD Ryzen Threadripper 3960X |
| GPU | NVidia TITAN Xp |
| RAM | 64 GB, 2666 MHz |

