# OpenReview forum: "Provably Accurate ODE Forecasting Through Explicit Trajectory Optimization"
_ICLR.cc/2024/Conference — Submitted to ICLR 2024_

### Official Review · Reviewer_U4Us · 2023-10-15

**Soundness:** 3 good
**Presentation:** 4 excellent
**Contribution:** 1 poor
**Rating:** 3
**Confidence:** 4

**Summary:**

The paper presents a method for forecasting time series governed by ordinary differential equations (ODEs), focusing on future trajectories. The authors showed that the solution space of such ODEs is a finite-dimensional Riemannian manifold, allowing for applying statistical objectives like maximum likelihood and minimum mean squared error directly in the feasible ODE solutions space. This approach aims to reduce forecasting errors compared to traditional methods not based on the solution manifold, further supported by numerical examples.

**Strengths:**

1. The paper showed the space of trajectories on a finite time interval I, with a finite degree of freedom, is a finite-dimensional manifold embedded in the space of a square-integrable function.

2. The paper summarized several classic estimation methods for trajectory forecasting and numerically compared their effects on the Lotka--Volterra equation.

3. The paper is very clearly written.

**Weaknesses:**

1. I find the paper presents an overly simple idea with an unrealistic assumption. When the space of parameters $\Theta$, the space for the inputs $\mathcal{U}$ are both finite-dimensional, finding the unknown dynamics is, of course, a finite-dimensional problem. However, most of the time,  the space of parameters $\Theta$/the input space $\mathcal{U}$ are not finite-dimensional. This is why forecasting dynamics is hard. For example, suppose the initial condition is not known exactly but on an open interval, which is no longer finite-dimensional. In that case, we do not know what type of chaotic behavior will occur in the long-time horizon. The same thing applies to the parameter space. General variable coefficients, even for the smoothest functions in $C^\infty$, are in an infinite-dimensional space before further assumptions.

2. Even if we settle for finite-dimensional spaces for the initial condition, the parameters, the inputs, etc. (i.e., all the places where one can change the dynamics), the overall dimension can be large. Indeed, $C_{f,I}$ is the range of the forward problem, containing all possible trajectories. It is not possible to characterize the space analytically, so it will have to be done through the finite number of parameters mentioned above. In that sense, (13) becomes the standard parameterized minimization problem, like those we see in regressions. If we know a priori that the dynamics are only subject to a finite number of parameters, no one will use (12), which does not utilize this prior/expert knowledge.

3. The numerical examples are simple, and the white noise assumption for the data, as shown in (17), is also too simple. The proposed method will perform poorly if the noise can be fit into the trajectory manifold $C_{f,I}$.

4. The paper titled "**Provably accurate** ODE forecasting through explicit trajectory optimization". I don't see where the "provably accurate" is based. Both Theorem 1 and Proposition 1 are results with little to do with ODE forecasting. Theorem 1 is a consequence of the "finite dimension" assumption, and Proposition 1 is a property from differential manifold.

**Questions:**

1. The metric tensor K is diagonal, in the sense that it is zero if the time points don't match. This is very special and does not accommodate correlations in time. Why do the authors want to choose this?

2. Is the norm $ || \cdot||$ in (12) the same as in (13)? If so, are you not giving the manifold a particular metric but using the one for $L^2(I)$?

3. It was $\hat{x}$ in (13), but became $\tilde{x}$ in (16)?

4. Figure 2 is a bit confusing. For some, we need to look for the maximum, while for some, we need to look for the minimum. It is better that they are all for maximum or all for minimum. Otherwise, it is confusing and hard to compare across different figures.

---

> ### Author Response · Authors · 2023-11-22
> **Response to U4Us, Part 1**
>
> > "I find the paper presents an overly simple idea with an unrealistic assumption. When the space of parameters $\Theta$, the space for the inputs $\mathcal{U}$ are both finite-dimensional, finding the unknown dynamics is, of course, a finite-dimensional problem. However, most of the time, the space of parameters $\Theta$/the input space $\mathcal{U}$ are not finite-dimensional. This is why forecasting dynamics is hard. For example, suppose the initial condition is not known exactly but on an open interval, which is no longer finite-dimensional. In that case, we do not know what type of chaotic behavior will occur in the long-time horizon. The same thing applies to the parameter space. General variable coefficients, even for the smoothest functions in $C^\infty$, are in an infinite-dimensional space before further assumptions."
>
> We suspect there may be some confusion in this concern regarding the meaning of finite-dimensional spaces.
> The example provided, "the initial condition is not known exactly but on an open interval," is one-dimensional as it is a continuous subset of the real line, a one-dimensional vector space.
> General variable coefficients are indeed allowed by the finite-dimensionality assumption.
>
> The finite-dimensionality assumption is commonly satisfied by many real-world systems such as robotics or circuits.
> These come from practical engineering constraints, where construction of arbitrary input functions is infeasible due to physical constraints such as damping in a system.
>
> > "Even if we settle for finite-dimensional spaces for the initial condition, the parameters, the inputs, etc. (i.e., all the places where one can change the dynamics), the overall dimension can be large. Indeed, $\mathcal{C}_{f,I}$ is the range of the forward problem, containing all possible trajectories. It is not possible to characterize the space analytically, so it will have to be done through the finite number of parameters mentioned above.
>
> It is true that the space of solutions cannot be characterized analytically, as even simple ODEs do not have analytical solutions, though the lack of existence of analytical solutions does not cause any issues with inference.
> A lack of analytical solutions does not prohibit the usage of the smooth structure in optimization problems, allowing gradient-based methods to be applied.
>
> > "In that sense, (13) becomes the standard parameterized minimization problem, like those we see in regressions. If we know a priori that the dynamics are only subject to a finite number of parameters, no one will use (12), which does not utilize this prior/expert knowledge."
>
> Equation (13) is *not* the current approach today, and we hope that other researchers begin to adopt it after reading this manuscript.
> We suspect this is because it was not previously known that the objective is well-behaved enough to solve.
> This guarantee comes from the core contribution of this work.
>
> Neural ODEs and related work (Chen et al. 2018; Dupont et al., 2019; etc.) fit a trajectory to the observations, similar to a classical regression method.
> In Bayesian statistics, Equation (12) represents the minimum mean squared error estimator, and is one of the main objectives of MCMC sampling.
>
> > "The numerical examples are simple, and the white noise assumption for the data, as shown in (17), is also too simple."
>
> We selected the Lotka-Volterra equation because it provides a clear illustration of the difference between the state estimation problem and the forecasting problem. As trajectories of the system form trains of short pulses with period dependent on the initial conditions, successful forecasting is highly dependent on predicting the locations of the pulses. This feature allows clear visualization of a trade-off in estimation: improve the accuracy of the shape of the initial pulse, or choose an initial condition that better captures the timings.
>
> There is no fundamental dependence on white noise in this work, and any stochastic observation process could be chosen.
> The core contribution of this work is the theoretical advancement and the guarantee for optimality, irrespective of the noise process.
>
>
> > "The proposed method will perform poorly if the noise can be fit into the trajectory manifold $\mathcal{C}_{f,I}$"
>
> Unless we are misunderstanding the claim, the proposed noise process would result in the forecasting problem being impossible in general, not just for the our technique.
> If the noise is designed such that different valid solutions of the underlying model are indistinguishable from only the observables (i.e. the distribution of observables is independent of the true process), then the problem would be impossible for any statistical method.

---

> ### Author Response · Authors · 2023-11-22
> **Response to U4Us, Part 2**
>
> > "The paper titled 'Provably accurate ODE forecasting through explicit trajectory optimization'. I don't see where the 'provably accurate' is based. Both Theorem 1 and Proposition 1 are results with little to do with ODE forecasting. Theorem 1 is a consequence of the 'finite dimension' assumption, and Proposition 1 is a property from differential manifold."
>
> Theorem 1 and Proposition 1 enable provably optimal forecasting within a model family.
> This contribution is distinct from other approaches for which the objective function is dependent explicitly on the parameters rather than on the future predictions.
>
> > "The metric tensor K is diagonal, in the sense that it is zero if the time points don't match. This is very special and does not accommodate correlations in time. Why do the authors want to choose this?"
>
> The choice of a diagonal metric tensor in Equation (6) does not represent a limitation to this work, and the more general form in Equation (5) is applicable.
> As stated in the manuscript, the diagonal structure in Equation (6) was chosen to enable a re-weighting of the importance of different time horizons, a common requirement in forecasting problems.
> It represents a useful example which later has a fundamental implication for Section 4.3, where it implies point-wise optimality for the MMSE estimation in the ambient space.

---

### Official Review · Reviewer_YW1N · 2023-10-31

**Soundness:** 2 fair
**Presentation:** 4 excellent
**Contribution:** 3 good
**Rating:** 5
**Confidence:** 3

**Summary:**

This paper discusses a new approach for the estimation of controlled ODE solutions based on the geometric properties of flow maps. The authors first provide an overview of the problem and present their main theoretical results. Then, they discuss how the theory translates into practical algorithms and finally show performance on a Lotka-Volterra ODE. Other ODEs are studied in the appendix. In the appendix, one can also find a detailed discussion of some interesting elements, such at tolerance and stepsize selection.

**Strengths:**

The paper is rigorous, very well-written, and interesting. For me, truly a joy to read. I learned a lot from this paper and find the idea very nice. I especially like the fact that the authors guide the reader through proofs and through the literature. This is very helpful. I am arguably not an expert on ODE estimation, but I worked on this a bit and find the contribution quite relevant for applications.

**Weaknesses:**

While I generally like the paper there are some points that need revision and better clarity.

1) Experiments: They work pretty well, but two things are not clear to me (a) assumptions for computation of $\det D\psi$ and (b) what is the input to the training pipeline. (a) From the appendix, it seems you compute $\det D\psi$ assuming access to the exact solver. From what I understood in the paper, it instead seems you have access only to noisy measurements. (b) this is related. seems from the paper you assume access to a single noisy trajectory, but I guess you actually use more than one. Can you make this more clear to me?

2) Comparisons with other methods: This is totally missing. Only variants of the proposed methods are discussed. While the discussion is still interesting, I would like the authors to show comparisons with other alternatives for solving the problem. In terms of Accuracy, Speed, Assumptions. I am totally happy to revise this score if you are able to show this.

Typos (minor):
- In Abb B, I would remind the reader of the definition of $\psi$. Also, in the proof of Lemma 1, $\forall x_0\in\mathcal{X}$, not $x$.
- After formula 9, I would perhaps explain better why "this potential concern is unfounded". I would also tone down the sentence.

**Questions:**

I have a question.

3) In the main paper assumptions, you have that $\mathcal{U}$ is a manifold. However, in App. B, you assume the inputs have a finite-sum structure. Hence, it seems to me the assumption on the input set is a bit stronger than what you claim. Am I right?

---

> ### Author Response · Authors · 2023-11-22
>
> > "Experiments: They work pretty well, but two things are not clear to me (a) assumptions for computation of and (b) what is the input to the training pipeline. (a) From the appendix, it seems you compute assuming access to the exact solver. From what I understood in the paper, it instead seems you have access only to noisy measurements. (b) this is related. seems from the paper you assume access to a single noisy trajectory, but I guess you actually use more than one. Can you make this more clear to me?"
>
> Part (a):
> We assume knowledge that true model exists in some known parameterized family of ODEs.
> We then use a numerical solver to evaluate the associated transformations in this model family.
> These transformations are deterministic functions of the model family, and are not dependent on the observations.
>
> Part (b):
> We observe a single noisy trajectory.
> This noisy trajectory induces the probability density on which we operate with the transformations from part (a).
>
>
> > "Comparisons with other methods: This is totally missing. Only variants of the proposed methods are discussed. While the discussion is still interesting, I would like the authors to show comparisons with other alternatives for solving the problem. In terms of Accuracy, Speed, Assumptions. I am totally happy to revise this score if you are able to show this."
>
> We included simulations for the methods that are most relevant to this work.
> The core contribution in this work is the recognition that statistical cost functions are well-behaved on the space of solutions of a differential equation.
> It is for this reason that we compared the optimal solutions to these costs to their counterparts in the state estimation problem.
>
> Methods with weaker assumptions are unable to capture long-term structure of the trajectory (as seen in Appendix A.2).
> Furthermore, other methods emphasizing speed with similar assumptions typically aim to construct approximate solutions to the state estimation problem.
> Thus, we did not include them as we felt a more thorough characterization of the difference between the optimal solutions would be more salient.
>
> > "In the main paper assumptions, you have that is a manifold. However, in App. B, you assume the inputs have a finite-sum structure. Hence, it seems to me the assumption on the input set is a bit stronger than what you claim. Am I right?"
>
> We appreciate your careful reading of the appendices.
> You are correct regarding the mismatch between the paragraph at the beginning of Appendix B and the claim.
>
> We will revise the paragraph to state that the parameter space can be augmented as the product manifold of $\mathcal{U}$ and $\Theta$, rather than attempting to illustrate the process concretely through a linear subspace.

---

> > ### Comment · Reviewer_YW1N · 2023-11-22
> > **Thanks**
> >
> > Thanks for the comments! I cannot see a revision of the manuscript posted, are you sure you included this?

---

### Official Review · Reviewer_Rdcm · 2023-10-31

**Soundness:** 3 good
**Presentation:** 2 fair
**Contribution:** 2 fair
**Rating:** 3
**Confidence:** 3

**Summary:**

This paper presents a framework for predicting the behavior of ordinary differential equations. The research is centered on the study of smooth Lipschitz dynamical systems over a finite time interval and establishes that the set of possible trajectories forms a finite-dimensional Riemannian manifold. Through the integration of established estimation techniques, such as maximum likelihood, maximum a posteriori, and minimum mean squared error estimation, the authors introduce methods for computing the best-estimated trajectories. The paper also delves into the properties and conducts numerical experiments to illustrate specific estimations.

**Strengths:**

- The model formulation is rigorous, and the theoretical proofs are robust.

- After introducing the abstract framework, the paper discusses common estimation objectives and presents practical methods for computing trajectory estimations.

**Weaknesses:**

- The commonalities and differences between the author's approach and related methods are not thoroughly discussed, as indicated in the first question.

- While the authors assert that using objectives explicitly dependent on the system's trajectory is superior to point estimation, there is a lack of theoretical guarantees provided to substantiate this claim formally.

**Questions:**

- The commonalities and differences between the author's approach and related work in Section 1.1 are not clearly delineated. For instance:
  - Is the method proposed in this paper a variant of the Neural ODE method?
  - What advantages does the proposed method offer over existing Neural ODE methods in the context of time-series forecasting problems?
  - The relevance of discussing regularization methods for training neural networks to solve differential equations is not entirely clear.

- When the parameter space $\Theta$ s derived from a neural network, is it possible to verify whether $f$ is a smooth function, satisfying assumption 1?

- Could you please provide the definition of the function $\varphi$ (in page 4, line 15) and the notation $\varphi^{\gamma_i}(x_0)$ (in page 5, line 24)?

- What is the meaning of the bracketed term $[D\varphi^{\gamma_i}]$ in the equation (14)?

- Could you offer a comparison of the computational complexity between the proposed methods for computing trajectory estimations and the estimations of the initial condition?

- Regarding Figure 2, it appears to be somewhat confusing.
  - Is the caption of each subfigure indicating the names of methods? How do they correspond to the six different forecasting objectives used in simulations?
  - Could you provide further insights into which surfaces of objective functions yield better results and which do not? Are these outcomes affected by changes in the initialization point?
  - For the statement 'the trajectory MSE illustrates a valley of initializations ... structure not captured by any competing technique',  it seems the State MSE also exhibits a valley structure. Could you elaborate on this observation?

---

> ### Author Response · Authors · 2023-11-22
> **Response to Rdcm, Part 1**
>
> > "Is the method proposed in this paper a variant of the Neural ODE method?"
>
> The proposed method is broadly applicable, both to Neural ODEs and more general system parameter estimation problems.
> The proposed method is best thought of as an improved cost function when the chosen model family is represented by differential equations, and can used as a replacement for the typical cost function.
>
> > "What advantages does the proposed method offer over existing Neural ODE methods in the context of time-series forecasting problems?"
>
> Our approach is guaranteed to be optimal in minimizing a chosen forecasting cost for a particular model family.
> Neural ODE methods typically use a likelihood approach as a surrogate objective.
> In doing so, competing methods fit parameters based only on the observation horizon, rather than the forecasting horizon.
>
> > "The relevance of discussing regularization methods for training neural networks to solve differential equations is not entirely clear."
>
> Our proposed method and existing regularization methods seek to resolve the same issue: generalization of a model to the out-of-sample regime.
> While regularization is commonly used to increase model stability and reduce overfitting to any given sample, our proposed method instead transforms the cost to explicitly be in the regime of interest.
>
> > "When the parameter space $\Theta$s derived from a neural network, is it possible to verify whether $f$ is a smooth function, satisfying assumption 1?"
>
> Yes, the smoothness property requires the usage of smooth activation functions.
> The smoothness assumption is common, and is additionally required in Neural ODEs and related work (Chen et al. 2018; Dupont et al., 2019; etc.).
>
> > "Could you please provide the definition of the function (in page 4, line 15) and the notation (in page 5, line 24)?"
>
> The collection of functions $\{\varphi^\tau\}$ advance time in the system.
> They are defined such that $\mathbf{x}_{\tau + t} = \varphi^\tau(\mathbf{x}_t)$ for all $t$.
>
> > "What is the meaning of the bracketed term in the Equation (14)?"
>
> Similar to Equation (7), the $D$ notation represents the matrix of partial derivatives in local coordinates.
> Thus,
> $$D\varphi^{\tau_i} = \begin{bmatrix}
> \rule[.5ex]{2.5ex}{0.5pt} & D\_{\mathbf{v}_1}|\_{\mathbf{x}_0} \varphi^{\tau_i}& \rule[.5ex]{2.5ex}{0.5pt} \\\\
> \rule[.5ex]{2.5ex}{0.5pt} & D\_{\mathbf{v}_2}|\_{\mathbf{x}_0} \varphi^{\tau_i} & \rule[.5ex]{2.5ex}{0.5pt} \\\\
>  & \vdots &
> \end{bmatrix},$$
> where $\{\mathbf{v}_i\}$ is a basis of the local coordinates on the state space around $\mathbf{x}_0$.
>
> > "Could you offer a comparison of the computational complexity between the proposed methods for computing trajectory estimations and the estimations of the initial condition?"
>
> We believe that such a comparison would only be informative for the particular pair of differential equation and ODE solver being investigated.
> The primary source of the difference is in requiring the solutions of the differential equation to be computed over the entire forecasting horizon rather than the observation horizon.
> The relative amount of time required is dependent on the dynamically chosen step sizes in the ODE solver in the observation regime and the forecasting regime.
> These step sizes are dependent on the ODE solver, the differential equation, and the initial conditions, making it difficult to generalize statements on the computational cost.
>
>
> > "Is the caption of each subfigure indicating the names of methods? How do they correspond to the six different forecasting objectives used in simulations?"
>
> The title of each subfigure is the function in the plot, e.g., "Likelihood" represents the likelihood function of the parameters, "State Posterior" represents the posterior distribution of the state variable, etc.
> We will rework the caption to improve clarity.

---

> ### Author Response · Authors · 2023-11-22
> **Response to Rdcm, Part 2**
>
> > "Could you provide further insights into which surfaces of objective functions yield better results and which do not?"
>
> The objective function relating directly to prediction performance should yield better results in prediction performance.
> Figure 2 was intended to illustrate the qualitative differences between costs in the space of solutions and costs in the space of initial conditions, demonstrating that the "valley of initialization which lead to similar trajectories" is not captured by other objectives.
>
> > "Are these outcomes affected by changes in the initialization point?"
>
> This work does not propose a specific optimization algorithm, but rather a new cost function dependent directly on the forecasting performance.
> As such, there is no initialization point in our proposed method.
>
> > "For the statement 'the trajectory MSE illustrates a valley of initializations ... structure not captured by any competing technique', it seems the State MSE also exhibits a valley structure. Could you elaborate on this observation?"
>
> The key part of the sentence was "which lead to similar trajectories along the interval."
> We are referring to the fact that the proposed cost function meaningfully associates similar forecasts, rather than similar initializations.
> The state MSE figure would suggest that a significant reduction in the initial prey population would be possible without sacrificing performance, whereas the trajectory MSE plot demonstrates this is not the case, and that the path is extremely sensitive to the initial prey population.

---

### Official Review · Reviewer_kKvz · 2023-11-01

**Soundness:** 3 good
**Presentation:** 3 good
**Contribution:** 1 poor
**Rating:** 3
**Confidence:** 4

**Summary:**

The paper investigates statistical estimation of finite-dimensional parameter via maximum likelihood (ML), maximum a posteriori (MAP), and minimum mean squared error (MMSE) in the space of feasible ODE solutions when ODE is known and the only thing to be estimated is the unknown initial condition. The main insight is based on the classical results on the existence and uniqueness of the solutions of an ODE with Lipschitz continuous vector field with a continuously differentiable derivative, in which case there exists diffeomorphism  between the (finite-dimensional) state space and manifold of trajectiories with a finite time-horizon. Using the diffeomorphism, estimation is seen as a supervised learning problem for which ML, MAP and MMSE estimation can be formulated and solved.

Using standard argument, this setting is directly extended to estimating unknown initial condition $x_0\in{\mathcal X}$, parameters $\theta\in\Theta$ and input functions $u\in{\mathcal U}$, whenever $\mathcal X$, $\Theta$ and $\mathcal U$ are finite dimensional.

One illustrative example of the Lotka-Voltera predator-pray system is presented.

**Strengths:**

Paper has a clear story-line. Statistical estimation of the ODE parameters from the observed trajectories is an important scientific problem.

**Weaknesses:**

While I find the story-line of the paper nice and useful for readers interested in learning dynamical systems, my overall impression is that the submission is this form is not appropriate for acceptance for the ICLR. The main reasons are the following:

1) __Theoretical aspect.__ Main results of the paper seam, at best, based on well-known arguments, if not already present is the same form in the existing literature. How the paper is presented, my impression is that novelty is highly overstated. Giving proper references for all arguments based on classical ODE theory is necessary.

2) __Methodological aspect.__  In related works many references study different problem - estimating dynamics when ODE is not known, which is not the problem that this paper studies. On the other hand, existing literature on the ML MAP and MMSE estimation of the initial conditions of a known ODE is not properly reviewed, and the novelty of the considered methodology lacks perspective.

3) __Experimental aspect.__ One toy ODE model is by far bellow ICLR standard. No broader context presented. Such discussion is insufficient to draw any kind of reliable conclusions. Additional material in the Appendix is minor and in part (Section A.2) obvious.


__Minor issues:__

1) $\varphi$ is nowhere properly defined
2) Bellow Eq. (6) $R^{K\times K}$ should read $R^{N\times N}$
3) In Eq. (8) inner product is in $L^2(I)$ not an RKHS. Proposition with an RKHS should be stated, at least in the Appendix.
4)  Eq. (12) should be rewritten to avoid confusion. What is written now is that the empirical estimate equals the regression function.

**Questions:**

When introducing kernel for the trajectories, due to the change of geometries between RKHS and  $L^2(I)$ spaces, the existence and the properties of the diffeomorphism should be at least commented. Also, can you please clarify when the optimization is done in the RKSH norm and when in $L^2$ norm. To me it seams that aspects of the statistical learning theory of kernel methods are not addressed properly. In particular, regression function in Eq. (12) may or may not belong to the RKHS defined by the kernel, and the minimisation is typically not done in $L^2$ but in RKSH.  In particular, can you please elaborate on "_MMSE trajectory estimate is optimal for any desired weighting of time horizons by the construction of Equation (6)._"

---

> ### Author Response · Authors · 2023-11-22
> **Response to kKvz, Part 1**
>
> > "Main results of the paper seam, at best, based on well-known arguments, if not already present is the same form in the existing literature. How the paper is presented, my impression is that novelty is highly overstated."
>
> To our knowledge, the provided structural result is novel.
> Other related results either require significantly stronger system constraints (e.g. Quotient manifold theorem (Lee, 2013)) or fail to provide sufficient structure for practical applications (e.g. non-Hausdorff manifold of solutions discussed by Souriau (1997)*. Without the Hausdorff structure, many common tools break down due to lack of uniqueness of limits).
>
> The novelty is in identifying the appropriate set of constraints to induce sufficient structure for statistical analysis.
> While remaining broadly applicable, the set of constraints enables the ability to directly optimize forecasts over the entire time horizon of interest.
>
> *: J.-M. Souriau, _Structure of Dynamical Systems: a symplectic view of physics._ Springer Science+Business Media, New York, NY, 1997.
>
>
> > "Giving proper references for all arguments based on classical ODE theory is necessary."
>
> We believe that we have included proper references.
> When appropriate throughout the manuscript, we include references to "Nonlinear Systems" by Hassan Khalil, a standard book on the analysis of nonlinear ODEs.
> When discussing geometric approaches, we additionally reference "Introduction to Smooth Manifolds" by John Lee.
> We are unclear as to which arguments you feel are missing references; we could additionally include a real analysis book for the epsilon-delta proofs.
>
> > "In related works many references study different problem - estimating dynamics when ODE is not known, which is not the problem that this paper studies."
>
> We are unclear as to which references you feel study fundamentally different problems, as our work relies on the same assumptions as neural ODEs and related work (Chen et al. 2018; Dupont et al., 2019; etc.), is most comparable to physics-informed ML (Raissi & Karniadakis, 2018;
> Raissi et al., 2019), and represents a change in cost function similar to regularization methods.
> Furthermore, we believe that hierarchical forecasting  (Rangapuram et al., 2021; 2023) is implicitly operating under a similar structural assumption.
>
> The problem of estimating dynamics and making predictions are intrinsically linked.
> The benefits of algorithms such as neural ODEs are in their predictive power, not in capturing a interpretable representation of some fundamental physical principle.
> Furthermore, we do not assume a fully known ODE, but rather that the ODE exists in some parameterized family of candidate models, the same underlying assumption in such a work.
>
> > "On the other hand, existing literature on the ML MAP and MMSE estimation of the initial conditions of a known ODE is not properly reviewed, and the novelty of the considered methodology lacks perspective."
>
> We do not believe that existing literature on ML, MAP, and MMSE estimation in dynamic systems addresses the problem of interest in this work.
> Existing literature on specific objective functions largely focus on computationally efficient algorithms and approximations.
>
> > "One toy ODE model is by far bellow ICLR standard. No broader context presented. Such discussion is insufficient to draw any kind of reliable conclusions."
>
> The reliable conclusion in the manuscript is in the theoretical foundations of the work, not the numerical aspect.
> The proposed analysis guarantees optimality of the forecast within the model family.
> Simulations are included largely to characterize situations in which the performance gap may be meaningful (e.g., large time horizons in Figure 3).
>
> > "Additional material in the Appendix is minor"
>
> We believe that we have included extensive additional material throughout the appendix.
> Appendix A includes simulations for three additional systems and two competing techniques.
> Appendix B and Appendix D include full proofs for the main results.
> Appendix C includes a review of key properties in Riemannian geometry.
> Appendix E provides some analysis on tolerance selection for the ODE solvers used in the work.
>
> > "in part (Section A.2) obvious."
>
> While we agree that Section A.2 is well-known to many in this field, in past publications comparisons with these models have been specifically requested by reviewers.
> Thus, we have included the additional figures for those for whom it is not obvious.

---

> ### Author Response · Authors · 2023-11-22
> **Response to kKvz, Part 2**
>
> > "When introducing kernel for the trajectories, due to the change of geometries between RKHS and $L^2(I)$ spaces, the existence and the properties of the diffeomorphism should be at least commented.  Also, can you please clarify when the optimization is done in the RKSH norm and when in $L^2$ norm. To me it seams that aspects of the statistical learning theory of kernel methods are not addressed properly. In particular, regression function in Eq. (12) may or may not belong to the RKHS defined by the kernel, and the minimisation is typically not done in $L^2$ but in RKSH."
>
> While it can be shown that the solutions of the ODE exist in an RKHS directly based on continuity of the evaluation functional, we do not change from $L^2(I)$ into an RKHS by introducing a linear operator.
> It is true that positive definite integral operators induce an RKHS by Mercer's theorem, but we do not require the RKHS structure and thus feel it would serve as a distraction from the main contributions.
> The core contribution in this work is to show that the space of solutions is a Riemannian manifold, not an arbitrary structure in an RKHS, nor $L^2(I)$.
> The embedding in $L^2(I)$ is used only to enable naturally occurring cost functions through a restriction onto the embedded manifold and to inherit a metric.
>
> > "In particular, can you please elaborate on 'MMSE trajectory estimate is optimal for any desired weighting of time horizons by the construction of Equation (6).'"
>
> The cost function in Equation (6) is constructed through the application of an invertible transformation to the solution space.
> As the MMSE estimator commutes with linear transformations, the result is invariant to the choice of integral kernel.
> The integral kernel determines the weighting of time horizons in the cost.
>
> We appreciate the feedback and will expand the line in the manuscript to make this point more explicit.

---

> > ### Comment · Reviewer_kKvz · 2023-11-23
> > **Acknowledgment of the rebuttal**
> >
> > I thank the authors for their reply. I have read it, but, due to short remaining time, I cannot provide detailed discussion at this point. However, I will consider the entire rebuttal to base my final opinion during AC-reviewers discussion period that is starting soon.

---

### Meta-Review · Area_Chair_mTvh · 2023-12-06

**Metareview:**

This paper studies parameter estimation for ODE dynamical systems under certain regularity assumptions.

All reviewers are in agreement that the paper, while well-written, should not be accepted, because it does not sufficiently contextualize its results relative to the classic numerical analysis of ODE problems, and does not provide sufficiently convincing experiments. The discussion period did not dispel these concerns.

**Justification For Why Not Higher Score:**

I full agree with the reviewers' concerns about novelty. The paper does a poor job of connecting to classic numerical analysis.

**Justification For Why Not Lower Score:**

N/A

---

### Decision · Program_Chairs · 2024-01-16

Reject